# ONLINE PSEUDO-AVERAGE SHIFTING ATTENTION(PASA) FOR ROBUST LOW-PRECISION LLM INFERENCE: ALGORITHMS, NUMERICAL ANALYSIS AND PERFORMANCE

## ABSTRACT

Attention computation remains a critical bottleneck for long-sequence inference in large models (e.g., long-text/video generation). To address this, we propose PASA, a fully low-precision algorithm that maintains mathematical equivalence to Flash Attention while enabling stable FP16 computation. PASA introduces two key innovations: (1) online pseudo-average shifting and (2) global recovery, which jointly prevent overflow and preserve numerical accuracy by dynamically adjusting attention score statistics. This approach significantly reduces memory bandwidth requirement and leverages low-precision compute units on AI accelerators (e.g. NPUs). We identify that numerical instability in attention stems from a *resonance* mechanism-phase alignment (or anti-alignment) between query and key matrices along the head dimension, which amplifies triggers overflow. Experiments on Qwen2-7B and Stable-Video-Diffusion demonstrate that PASA eliminates these issues while achieving $1.2 \times -1.65\times$ speedup with fully FP16 reaching up to 170 TFLOPs/s on Ascend NPUs compared to the highly optimized vendor libraries. To our knowledge, this is the first work to enable fully FP16 acceleration for long-sequence attention (MHA/MQA) with guaranteed numerical stability.

## 1 INTRODUCTION

The inference efficiency of large language models (LLMs) is severely limited by the quadratic complexity ($O(N^2)$) of attention computation in Transformer architectures(Vaswani, 2017). This bottleneck becomes critical as long-context language and multi-modal models demand processing of extremely long sequences(Liu et al., 2024; Lin et al., 2024; Jacobs et al., 2023; Bai et al., 2023b). Furthermore, on modern accelerators(Jack, 2020; 2022; Ajay & Raymond, 2024; Heng et al., 2019) with hierarchical memory, inefficient data movement and limited memory space during attention operations further degrades latency and hardware utilization. Optimizing attention is therefore essential to practical LLM deployment.

FlashAttention (FA) (Dao et al., 2022b; Tri, 2024; Shah et al., 2024) has emerged as a dominant approach for accelerating attention computation. It employs online and tiling strategies to minimize memory accesses while preserving mathematical equivalence with standard attention. Alternative approaches leverage attention sparsity or linear approximations (Dao et al., 2022a; Gu & Dao, 2023) to achieve $O(N)$ complexity. However, FA remains prevalent in production systems (e.g., ChatGPT, DeepSeek, LLaMA) due to its reliability and generalizability. Modern FA implementations rely on three general aspects of optimizations: (1) hardware-aware kernel fusion, (2) pipelining algorithms and (3) low-bit quantization (e.g., INT8/INT4 or FP8/FP6/FP4 quantization). For instance, FlashAttention-3 (Shah et al., 2024) achieves 740 TFLOPS (FP16) and 1.2 PFLOPS (FP8) on H100 GPUs with these techniques. Nevertheless, to maintain the numerical stability and accuracy, the intermediate variables in FA are still dominated by FP32 which could be a performance bottleneck. More exactly, FA3.0 is only related to reducing the underflow error in the calculation of the attention matrix score. Recent studies reveal another accuracy bottleneck: non-zero mean values in attention key tensors introduce systematic biases that amplify attention computation errors (Zhang et al., 2024). This phenomenon is aligned with the backward error analysis principles (Wilkinson, 1965; Higham,

2002), where accumulation errors increase with the magnitude of the input bias. SageAttention (Zhang et al., 2024) proposes global bias subtraction to mitigate this issue, enabling safe FP8 quantization for multi-modal models. However, global bias correction conflicts with hardware pipelining and fusion optimizations for long sequences and may further introduce additional accumulation errors (Blanchard et al., 2020).

We put forward PASA (Pseudo-Average Shifting Attention), an online algorithm for low-precision attention computation with enhanced numerical stability and accuracy. Our key innovation lies in leveraging the translation invariance of softmax (Blanchard et al., 2021) through online block-wise processing. Unlike prior work (Zhang et al., 2024; Shah et al., 2024), PASA performs bias correction at the block level, achieving three advantages: (1) compatibility with hardware pipelining architectures, (2) reduced numerical error and overflow via local pseudo-average shifting, and (3) efficient implementation through batched matrix multiplications that fully utilize matrix units. Crucially, PASA maintains backward compatibility with FA's optimization strategies, enabling seamless integration with existing infrastructure. We evaluate PASA on both synthetic benchmarks and real-world LLMs/video generation models, demonstrating significant improvements in numerical accuracy, stability and hardware efficiency. Our work provides new insights into co-designing numerical algorithms with modern AI accelerator architectures.

## 1.1 FLASH ATTENTION

In attention calculation, the basic inputs include query, key and value matrices from embedding results. These three matrices are represented by $\mathbf{Q} \in \mathbb{R}^{B \times N \times S_1 \times d}$, $\mathbf{K} \in \mathbb{R}^{B \times N \times S_2 \times d}$ and $\mathbf{V} \in \mathbb{R}^{B \times N \times S_2 \times d}$. $B$ and $N$ denote the batch size and head number in multi-head attention(MHA), respectively. The size $S_1$ and $S_2$ are input sequence lengths. For self-attention whose query, key and value matrices are from same input tokens, so, $S_1 = S_2$. Besides, $d$ denotes the head dimension of the input matrices.

The standard attention calculation in the prefilling phase follows the four steps: (1) first matmul - $\mathbf{S} = \mathbf{Q}\mathbf{K}^T \in \mathbb{R}^{B \times N \times S_1 \times S_2}$; (2) static scaling - $\mathbf{S} = \mathbf{S}/\alpha \in \mathbb{R}^{B \times N \times S_1 \times S_2}$; (3) softmax - $\mathbf{P} = softmax(\mathbf{S}) \in \mathbb{R}^{B \times N \times S_1 \times S_2}$; (4) second matmul - $\mathbf{O} = \mathbf{P}\mathbf{V} \in \mathbb{R}^{B \times N \times S_1 \times d}$. The $\alpha = 1/\sqrt{d}$ is the static scaling factor. The naive implementation of the attention is not hardware-efficient and scalable. By introducing blocked algorithms to the above four steps, FA 1.0(Dao et al., 2022b), 2.0(Tri, 2024) and 3.0(Shah et al., 2024) have been put forward to the above problems. For the sake of brevity, the FA procedure is given in Section A.

## 1.2 PRECISION ALLOCATION STRATEGIES IN FLASH ATTENTION OPTIMIZATION

Thanks to high flops and throughput of specialized matrix units (e.g., NPU CUBE, GPU Tensor Cores), Flash Attention (FA) achieves several-fold speedups over standard attention with low-precision acceleration. In the original FA 1.0(Dao et al., 2022b) and FA 2.0(Tri, 2024), only the inputs to matrix units use FP16/BF16; all subsequent operations (matmul, scaling, softmax, online update) remain in FP32 (Figure 1), ensuring numerical stability and avoiding overflow with FP32's wide dynamic range.

In FP3.0(Shah et al., 2024), FP8 quantization is integrated into the FA 2.0 pipeline to further boost performance and reduce memory footprint. On GPUs and NPUs, data movement between different memory hierarchies is a dominant cost. FP32 allocations render FA memory-bound, undermining pipeline efficiency. Thus, a mixed-precision allocation scheme (Figure 2) casts the attention scores matrix $\mathbf{S}$ in FP16, as implemented in the Ascend/CANN library's `fused_infer_attention_score` operator, which offers both "high-performance" and "high-precision" PFA(prefill FA) modes. On Ascend NPU, using the high-performance variant yields up to a $1.7\times$ speedup by alleviating memory-movement bottlenecks. Extending this idea, a fully low-precision FA (Figure 3) executes nearly all tensors and operations in FP16, promising further gains in performance and energy efficiency.

However, FP16's reduced exponent range (see Table 2) increases the risk of underflow and overflow (`INF` values), especially in diverse prompts and multi-modal inference. Consequently, directly adopting the mixed- or full-FP16 allocation schemes (Figures 2 and 3) may compromise

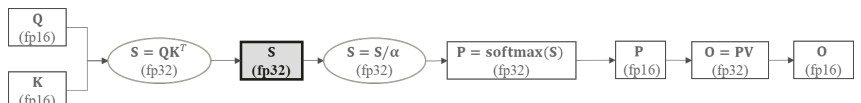

Figure 1: Precision allocation in original FA (1.0/2.0).

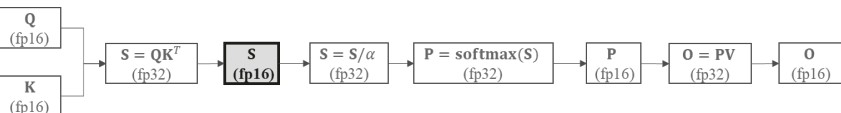

Figure 2: Mixed-precision FA: FP16 for attention score matrix.

numerical robustness. In the next section, we introduce PASA, a enhanced robust FA algorithm that preserves numerical stability and accuracy under full-FP16 for LM inference.

## 2 PASA: Pseudo-average Shifting Attention

The framework of PASA is compatible with the original FA, which means that they are mathematically equivalent if not considering numerical rounding error. However, the origin of numerical overflow is eliminated in PASA by adding some mathematical operations compared to the original FA. To introduce PASA algorithm, it is essential to firstly present the possible overflow positions in the original FA.

### 2.1 The Possibility Analysis of the Appearance of Large Numbers in FA

Assuming $\mathbf{Q}, \mathbf{K} \mathbf{V}$ lie within FP16's representable range (e.g., $|x| \leq 65504$), overflow can only occur between Equation (12) and Equation (19). In Equation (12), the GEMM inner product may amplify the maximum norm: $\|\mathbf{S}\|_{max} >> \|\mathbf{Q}\|_{max}, \|\mathbf{K}\|_{max}$. By contrast, the static scaling in Equation (13) attenuates values by a factor $\alpha > 1$. The softmax step (Equation (14), Equation (15)-Equation (17)) cannot increase magnitude: the max-reduction followed by $\exp(S_i^j - m_i)$ with $S_i^j - m_i \leq 0$ ensures $\mathbf{P} \ll 1$. Likewise, the block-wise sum operations in Equation (17) and Equation (18), performed on blocks of size $\leq 256$, prevent overflow. In summary, the initial GEMM in Equation (12) is the primary source of potential overflow in FA.

### 2.2 The Mathematical Framework of PASA

It is well known that softmax is translation-invariant(Blanchard et al., 2021), a property FA exploits by subtracting the row maximum in Equation (14). Equivalently, one may pre-shift the key matrix:

$$softmax(\mathbf{Q}\mathbf{K}^T) = softmax(\mathbf{Q}(\mathbf{K}^T - \mathbf{K}_0^T)) \tag{1}$$

where $\mathbf{K}_0 = [\mathbf{k}; \ldots; \mathbf{k}]$ contains identical bias vectors $\mathbf{k}$ with $\mathbf{k} \in \mathbb{R}^{1 \times d}$.

SageAttention(Zhang et al., 2024) sets $\mathbf{k}$ to the whole-sequence mean of $\mathbf{K}$, demonstrating that a large shared bias in $\mathbf{K}$ amplifies numerical error in attention. PASA similarly applies Equation (1) but computes $\mathbf{K}_0$ in a local online manner as a pseudo-average, enhancing locality and reducing numerical error as well as overflow risk in long sequences. We analyze this bias-induced overflow in the experiments of Section 3.3.1.

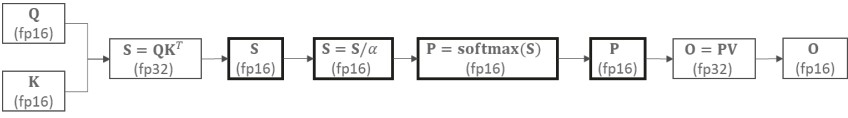

Figure 3: Full low-precision FA: FP16 for all operations.

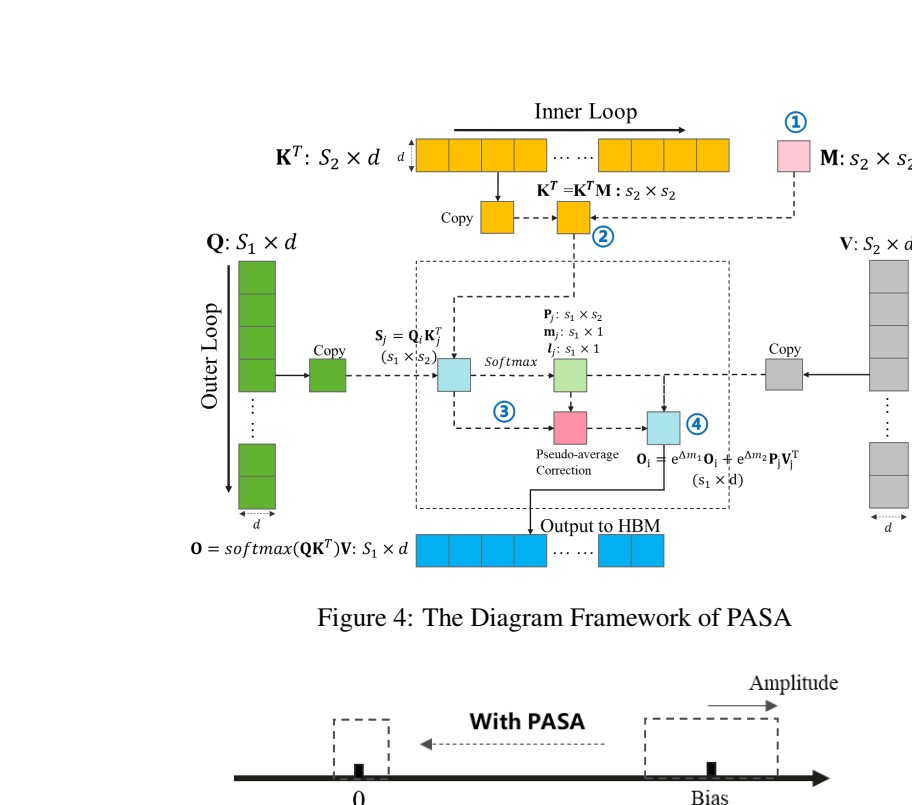

Figure 4: The Diagram Framework of PASA

Figure 5: The Diagram for the Reduction of both Average Value and range with PASA.

Moreover, PASA is able to integrate seamlessly with existing FA pipelines, including block and online quantization(Shah et al., 2024) and distributed Ring Attention(Liu et al., 2023). Its framework (Figure 4) and hardware-friendly algorithm (Algorithm 1) reduces to FA 2.0 when the hyperparameter $\beta = 0$. When inputs $\mathbf{Q}, \mathbf{KV}$ use BF16, PASA casts them to FP16 to preserve accuracy. The core workflow mirrors FA 2.0, with four additional steps denoted ①②③④.

### 2.2.1 CONSTRUCTION OF THE SHIFTING MATRIX AND MATRIX PREPROCESSING FOR STEPS ① AND ②

The naive bias-subtraction on $\mathbf{K}$ involves a reduction followed by a per-row subtraction, typically executed on low-throughput vector cores. Moreover, long-sequence inference accumulates significant rounding error. To address this, PASA implements a matrix-level approach-suitable for NPU CUBE or GPU Tensor Cores-via a single "shifting" matrix:

$$\mathbf{M} = \frac{\mathbf{I}}{\alpha} - \frac{\beta \mathbf{J}}{\alpha s_2} = \frac{1}{\alpha} \begin{bmatrix} 1 - \beta/s_2 & \dots & -\beta/s_2 \\ \vdots & \ddots & \vdots \\ -\beta/s_2 & \dots & 1 - \beta/s_2 \end{bmatrix} \tag{2}$$

where $\alpha = \sqrt{d}$ is the FA scaling factor. $\beta$ (a hyperparameter) controls the fraction of the mean key bias to subtract. Applying $\mathbf{M}$ right-multiplies any matrix by subtracting its pseudo-average implicitly.

By merging pseudo-average computation, bias subtraction, and static scaling into one GEMM-friendly step, PASA markedly reduces both the mean and range of $\mathbf{S}$ (see Figure 5), lowering overhead and adapting to transformers with varied bias magnitudes. Concretely,

$$\mathbf{K}_j'^T = \mathbf{K}_j^T \mathbf{M} = (\mathbf{K}_j^T - \beta \bar{\mathbf{K}}_j^T)/\beta \in \mathbb{R}^{d \times s_2} \tag{3}$$

$$\mathbf{S}_i'^j = \mathbf{Q}_i \mathbf{K}_j'^T = (\mathbf{Q}_i \mathbf{K}_j^T)\mathbf{M} = \mathbf{S}_i^j \mathbf{M} \in \mathbb{R}^{s_1 \times s_2} \tag{4}$$

Thus, subtracting $\mathbf{K}$'s bias is equivalent to subtracting from the attention scores $\mathbf{S}$, eliminating the root source of large values in FA.

---

**Algorithm 1** PASA Algorithm

---

1: **Input:** Matrices $\mathbf{Q}, \mathbf{K}, \mathbf{V}$ (Precision=`tp`) and transformation matrix $\mathbf{M}$ (Precision=`FP16`). Here, $\mathbf{Q} \in \mathbb{R}^{S_1 \times d}, \mathbf{KV} \in \mathbb{R}^{S_2 \times d}, \mathbf{M} \in \mathbb{R}^{s_2 \times s_2}$, and the sub-block matrices of $\mathbf{Q}$ and $\mathbf{KV}$ are $\mathbf{Q}_i \in \mathbb{R}^{s_1 \times d}, \mathbf{KV}_j \in \mathbb{R}^{s_2 \times d}$, while sub-block number are $N_q = S_1/s_1$ and $N_{KV} = S_2/s_2$ with the hyper-parameter - $\beta$ determined by optimal accuracy condition.

2: **Output:** the output $\mathbf{O} \in \mathbb{R}^{S_1 \times d} = [\mathbf{O}_1,, \mathbf{O}_{N_q}]$ with the sub-block matrix $\mathbf{O}_i \in \mathbb{R}^{s_1 \times d}$.

3: Construct the Shifting Matrix - $\mathbf{M}$ for the Treatment of Matrix - $\mathbf{K}$.

4: Initialize the Maximum vector: $m_0 = \mathbf{0} \in \mathbb{R}^{s_1 \times 1}, l_0 = \mathbf{0} \in \mathbb{R}^{s_1 \times 1}$. (Precision=`FP16`)

5: **for** $j = 1$ **to** $N_{KV}$ **do**

6:     (Batched-GEMM)Pre-processing: $\mathbf{K}_j^T = \mathbf{K}_j^T \mathbf{M} \in \mathbb{R}^{d \times s_2}$. (Precision=`FP16`)

7: **end for**

8: **for** $i = 1$ **to** $N_q$ **do**

9:     Initialize the output $\mathbf{O}_i = \mathbf{0} \in \mathbb{R}^{s_1 \times d}, l_0, m_0, \Delta m_0' = \mathbf{0} \in \mathbb{R}^{s_1 \times 1}$. (Precision=`FP16`)

10:     **for** $j = 1$ **to** $N_{KV}$ **do**

11:         (GEMM)Compute Attention Matrix: $\mathbf{S}_i^j = \mathbf{Q}_i \mathbf{K}_j^T \in \mathbb{R}^{s_1 \times s_2}$. (Precision=`FP16`)

12:         Compute Softmax: $m_j' = rowmax(\mathbf{S}_i^j) \in \mathbb{R}^{s_1 \times 1}, \mathbf{P}_i^j = exp(\mathbf{S}_i^j - m_j') \in \mathbb{R}^{s_1 \times s_2}, l_j' = rowsum(\mathbf{P}_i^j) \in \mathbb{R}^{s_1 \times 1}$. (Precision=`FP16`)

13:         Compute Pseudo-average Value: $\bar{\mathbf{S}}_i'^j = rowmean(\mathbf{S}_i^j) \in \mathbb{R}^{s_1 \times 1}$. (Precision=`FP16`)

14:         Compute Global pseudo-average Value: $\bar{\mathbf{F}}^j = \frac{(j-1)\bar{\mathbf{F}}^{j-1} + \bar{\mathbf{S}}_i'^j}{j} \in \mathbb{R}^{s_1 \times 1}$. (Precision=`FP16`)

15:         Correction Terms for $m$: $\Delta m_{j-1}' = \frac{\beta(\bar{\mathbf{F}}^{j-1} - \bar{\mathbf{F}}^j)}{1-\beta}, \Delta m_j' = \frac{\beta(\bar{\mathbf{S}}_i'^j - \bar{\mathbf{F}}^j)}{1-\beta}$. (Precision=`FP16`)

16:         Update $m$: $m_j = max(m_{j-1} + \Delta m_{j-1}', m_j' + \Delta m_j') \in \mathbb{R}^{s_1 \times 1}$. (Precision=`FP16`)

17:         Residual: $\Delta m_{j-1} = m_{j-1} - m_j + \Delta m_{j-1}', \Delta m_j = m_j' - m_j + \Delta m_j'$. (Precision=`FP16`)

18:         Update $l$: $l_j = exp(\Delta m_{j-1})l_{j-1} + exp(\Delta m_j)l_j' \in \mathbb{R}^{s_1 \times 1}$. (Precision=`FP16`)

19:         (GEMM)Compute Temporary Output: $\mathbf{O}_i^j = \mathbf{P}_i^j \mathbf{V}_j$. (Precision=`FP16`)

20:         Update the Output: $\mathbf{O}_i^j = exp(\Delta m_j)\mathbf{O}_i^j + exp(\Delta m_{j-1})\mathbf{O}_i^{j-1} \in \mathbb{R}^{s_1 \times d}$. (Precision=`FP16`)

21:     **end for**

22:     Update the Final Output: $\mathbf{O}_i = \mathbf{O}_i^{N_{KV}}/l_{N_{KV}}$. (Precision=`FP16`)

23: **end for**

24: Return the Output $\mathbf{O}$.

---

### 2.2.2 ONLINE RECOVERING OF GLOBAL BIAS INFORMATION FOR STEP ③

The above local block-wise subtraction of the pseudo-average bias implies that each block uses a different shift, so their maxima and softmax normalizations are not directly comparable.

For the current $i$-th row and block $(i, j)$, we first compute the uncorrected maxima $m_j'$ and row-sums $l_j'$, as well as the softmax weights $\mathbf{P}_i^j$ via Equation (14) on $\mathbf{S}_i^j$. Since $(m_j', l_j', \mathbf{P}_i^j)$ cannot be compared to the corresponding values from block $(i, j-1)$, we must apply global correction terms before performing the max-reduction in Equation (15). The following theorem underlies our online recovery of the pseudo-average bias.

**Theorem 1.** *Let $\mathbf{I} \in \mathbb{R}^{s \times s}$ represents the identity matrix, and $\mathbf{J} \in \mathbb{R}^{s \times s}$ is the all-ones matrix. $\lambda$ is a parameter not smaller than zero. The shifting matrix is defined as $\mathbf{M} = \mathbf{I} - \lambda \mathbf{J}$. The inverse of the shifting matrix is $\mathbf{M}^{-1} = \mathbf{I} + \frac{\lambda}{1-\lambda s}\mathbf{J}$, where the necessary and sufficient condition of the existence of the inverse is that $\lambda \neq 1$.*

Applying Theorem 1 to the shifted attention scores yields

$$\mathbf{S}_i'^j \left(\mathbf{I} + \frac{\beta}{(1-\beta)s_2}\mathbf{J}\right) = \mathbf{S}_i^j, \tag{5}$$

from which the per-row averages satisfy

$$\frac{\bar{\mathbf{S}}_i'^j}{1-\beta} = \bar{\mathbf{S}}_i^j \in \mathbb{R}^{s_1 \times 1}, \quad \bar{\mathbf{S}}_i'^j = \frac{1}{s_2}\text{rowsum}(\mathbf{S}_i'^j), \ \bar{\mathbf{S}}_i^j = \frac{1}{s_2}\text{rowsum}(\mathbf{S}_i^j). \tag{6}$$

We then maintain a running global average across blocks:

$$\bar{\mathbf{F}}^j = \frac{(j-1)\,\bar{\mathbf{F}}^{j-1} + \bar{\mathbf{S}}_i'^{j}}{j} \in \mathbb{R}^{s_1 \times 1}, \quad \bar{\mathbf{F}}^1 = \bar{\mathbf{S}}_i'^{1}, \tag{7}$$

where $\bar{\mathbf{F}}^j$ aggregates the first $j$ blocks. Using $\bar{\mathbf{F}}^{j-1}$ and $\bar{\mathbf{F}}^j$, we compute the correction terms $\Delta m'_{j-1}$ and $\Delta m'_j$ in Algorithm 1, then update the global maximum $m_j$ (and its correction $\Delta m_{j-1}, \Delta m_j$) across blocks $(i, j-1)$ and $(i, j)$.

### 2.2.3 Online Correction of the Softmax and Output for Step ④

After obtaining the correction terms, the following softmax summation in the $j-th$ block can be corrected to obtain $l_j$. Similarly, the output for the $i-th$ block is also able to be updated to obtain $\mathbf{O}_i$. When the $i-th$ row is finished, the final output $\mathbf{O}_i$ is updated by using the Equation (19).

### 2.3 Optimal Accuracy Condition of PASA in Low Precision Computing

When using finite-precision (e.g., FP16), PASA's mathematical equivalence to standard FA no longer holds, so we must account for rounding error and select $\beta$ to maximize accuracy. We define the optimal-accuracy condition as

$$\frac{\beta}{1-\beta} = f(\beta), \tag{8}$$

where $f(\beta)$ is the nonlinear function given in Equation (20) (Section C). Its derivation appears in Section D. We solve Equation (8) by fixed-point iteration (Equation (22)) in high precision (FP64), initializing $\beta$ as $1 - 2^{-4}$, $1 - 2^{-5}$, and $1 - 2^{-6}$. The resulting solutions are $\beta = 0.937500,\ 0.968994,\ \text{and}\ 0.984497$, respectively. In our experiments, we use $\beta = 0.984497$ for validation.

## 3 Experimental Validation

The validation for PASA algorithm is divided into numerical stability of avoiding overflow, numerical accuracy and performance on Ascend NPU. Datasets from both synthetic benchmarks and real large models of language and video are considered.

### 3.1 Benchmark Cases

To include the influence of outlier in the recently published FA 3.0(Shah et al., 2024), both the uniform random distribution and the hybrid random distribution are considered as depicted in Table 1. The random cases are generated using Equation (9) and Equation (10).

$$\mathbf{Q}, \mathbf{K}, \mathbf{V} = U(x_0 - Am, x_0 + Am), \tag{9}$$

$$\mathbf{Q}, \mathbf{K}, \mathbf{V} = N(x_0, 1) + N(0, Am^2) * Bernoulli(p) \tag{10}$$

where $U(a, b)$ represents uniform random distribution between the range of $[a = x_0 - Am, b = x_0 + Am]$ for the mean value and the range as $x_0$ and $Am$, separately. $N(\mu, \sigma^2)$ is the normal random distribution with the mean value and standard deviation as $\mu = x_0$ and $\sigma = Am$, respectively. Besides, the term - $Bernoulli(p)$ represents the Bernoulli distribution with the probability as $p = 0.001$ same as the setup in the literature (Shah et al., 2024).

The metic to measure the effect of avoiding overflow is to observe the `INF` or `NAN` in attention results. When it comes to numerical accuracy, we used the relative root-mean-square error(RMSE) in Equation (11). For real large models in Table 1, the inference results are compared with the reference.

$$RMSE = \|O_{computed} - O_{Golden}\|_2 / \|O_{Golden}\|_2 \tag{11}$$

### 3.2 Validation Platform

Experiments were conducted on Huawei's Ascend NPU(Ascend, 2022) with uncoupled CUBE and Vector units, a representative domain-specific architecture (DSA) for AI acceleration. The more

Table 1: Validation Benchmark Cases for PASA: Random Cases and Real Large Models(Qwen(Bai et al., 2023a; Yang et al., 2024) and Stable-Video-Diffusion(Blattmann et al., 2023)).

| Datasets | Category |
|---|---|
| Uniform Random | Random |
| Hybrid Random | Random |
| Qwen2-7B | Language |
| stable-video-diffusion-img2vid | Multi-modal |

details about NPU with uncoupled architecture are given in Section F. We use the heterogeneous Compute Architecture for Neural Networks (CANN) framework(Ascend, 2024), which provides the driver, runtime, the Ascend C programming model, and AI operator libraries. We compare PASA against CANN's high-precision mode of `fused_infer_attention_score` (hereafter "CANN-PFA") in CANN Operator Library v8.0.0. All tests run under the Torch-NPU framework, which extends standard PyTorch with Ascend NPU support. Our accuracy and stability testing code, written in Python and Torch-NPU, will be open-source.

### 3.3 NUMERICAL ACCURACY VALIDATION

For numerical accuracy and stability experiments, we keep the input shape of matrices - $\mathbf{Q}$ and $\mathbf{KV}$ as $(B, N, S, D) = (1, 16, 1280, 128)$ similar to FA3.0(Shah et al., 2024) for random datasets. For real LM cases in Qwen2-7B(Bai et al., 2023a; Yang et al., 2024) and Stable-Video-Diffusion(Blattmann et al., 2023), the input shapes are determined by LM network structures and context/video size.

#### 3.3.1 CASE 1: BENCHMARK DATASETS FOR RANDOM DISTRIBUTION

As noted in Section 2.2 and Figure 5, PASA reduces both the mean bias and the range of the attention score matrix. To investigate its effects on numerical error and overflow, we generate random inputs with varying means and ranges. Section G summarizes results for two distributions: uniform (Equation (9)) and hybrid (Equation (10)). Detailed numerical behaviors are also described there; here, we highlight the key findings for the uniform case:

1. Both large mean values and ranges can trigger overflow when FA is executed in FP16.
2. PASA consistently outperforms the partially low-precision FA(also high-performance PFA in CANN) scheme in overall numerical accuracy.

We further report the fraction of `NaN` outputs in Table 4 (Section H). The cases with a large uniform mean of $x_0 = 30$ and small range of $0.5$ yield entirely `NaN` attention outputs under partially low-precision FA as well as fully low-precision FA without PASA. By contrast, the cases with a large range and a smaller uniform mean of 20 suggest a smaller `NaN` rates to only $0.12\%$-$8.14\%$. Nevertheless, these levels still cause inference failures in LMs.

Finally, hybrid-distribution tests mirror the uniform-case results: large means and ranges in $\mathbf{Q}, \mathbf{K}$ drive overflow or `NaN` under low-precision FA. To assess real-model behavior, we perform analogous experiments on actual LMs, as presented in the next section.

#### 3.3.2 CASE 2: REAL LARGE MODELS

To verify the overflow mechanism in real-world large models, we conducted experiments on representative open-source models, as shown in Table 1. We identified overflow in the attention calculation by checking whether the matrix multiplication result $\mathbf{QK}^T$ exceeds the maximum normal value (65504) in `FP16` precision. In the overflow cases we observed, the shapes of the Query, Key, and Value tensors were $[B, H, S, D] = [1, 28, 5676, 128]$ for the Qwen2-7B model and $[50, 5, 9216, 64]$ for the IMG2VID model. In Figure 13(a-b) and Figure 14(a-b) of Section I, we plot cloud maps of the $\mathbf{QK}^T$ matrices. The original $\mathbf{QK}^T$ exhibits intense oscillations along the head dimension and a pronounced bias along the sequence dimension. After preprocessing with PASA, however, the data ranges shrink dramatically: for the Key matrices, from $[-412.0, 234.0]$ to $[-12.54, 9.976]$ in Qwen2-7B, and from

$[-34.44, 33.88]$ to $[-4.283, 5.843]$ in IMG2VID. Consequently, the attention-score ranges reduce from $[-226360, 27757]$ to $[-58134, 1124]$ and from $[-86569, -67503]$ to $[-3402, 1752]$ for the two models, respectively-ranges that `FP16` can represent without overflow.

Moreover, we plot the sampling data along the sequence dimension for the original $\mathbf{QK}^T$ matrices from IMG2VID in Figure 7(a), alongside the PASA-preprocessed results in Figure 7(b). Similar phenomena are observed for Qwen2-7B in Section J. Hence, we draw two key findings:

1. The `resonance` between Query and Key matrices along the head dimension causes the large values in the attention scores for both the Qwen2-7B (language) and IMG2VID (video) models;

2. PASA effectively suppresses the origin of these large values by eliminating the resonance range along the head dimension.

We define `resonance` by analogy to physics as the extreme amplification of range when the frequency (or wavelength) and phase of an external force match those of a system. In our attention calculations, the Query matrices act as the external forces. `Resonance` occurs when the frequency (along head dimension) of the Query matches that of the Key, with a phase lag of $180°$, producing large negative inner-product values in $\mathbf{QK}^T$ and risking overflow in half precision. Hence, we formalize this in Figure 6, distinguishing two cases:

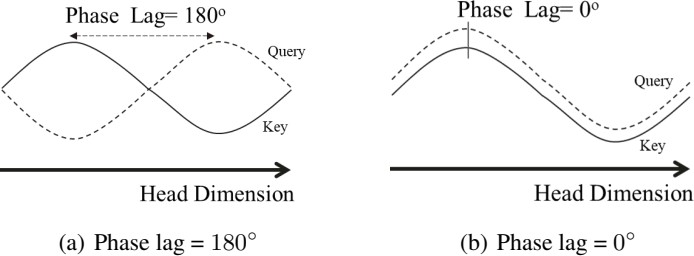

(a) Phase lag = $180°$         (b) Phase lag = $0°$

Figure 6: Definition of `Resonance` in Attention Calculation. Case (a) causes large negative values; case (b) causes large positive values.

Finally, to demonstrate PASA's end-to-end effect on inference accuracy, we show two generated video clips from IMG2VID in Figure 8 (input prompts in Section L). No overflow occurs during inference with PASA, and the generated video quality matches the reference. Besides, same inference results hold for long-text tasks in Qwen2-7B (see Section L). This confirms that full-FP16 attention with PASA can support high-quality, numerically robust inference without sacrificing accuracy.

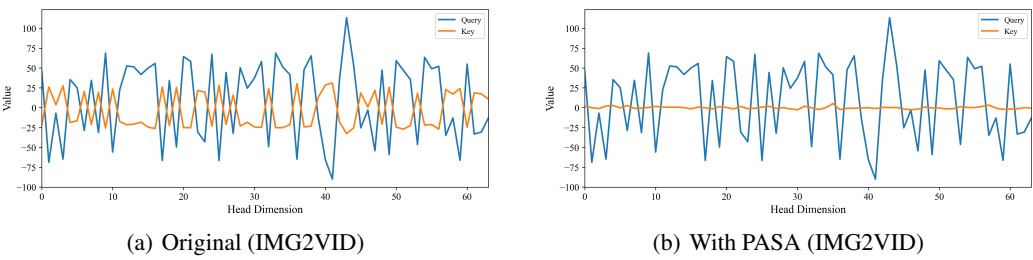

(a) Original (IMG2VID)         (b) With PASA (IMG2VID)

Figure 7: Sampling Data Distributions for Query/Key along Head Dimension Direction(IMG2VID).

### 3.4 PERFORMANCE

To validate the performance gains resulting from the full-FP16 precision allocations for attention computation using PASA, we conducted scaling experiments across various long sequence lengths and compared against the high-precision implementation of CANN-PFA. Both implementations are

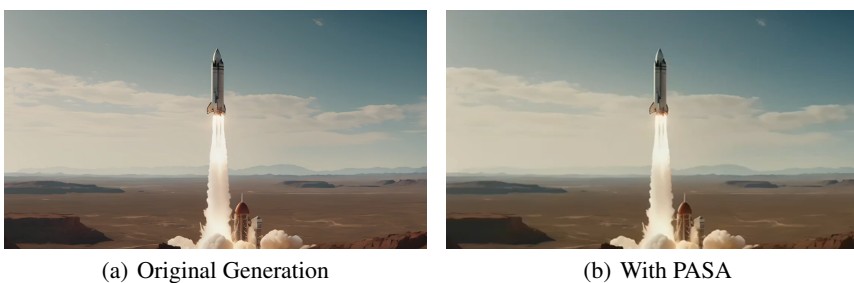

(a) Original Generation         (b) With PASA

Figure 8: Generated video frames from IMG2VID without and with PASA(Structural Similarity (SSIM): 0.9847 (SSIM > 0.95 indicates high quality match)).

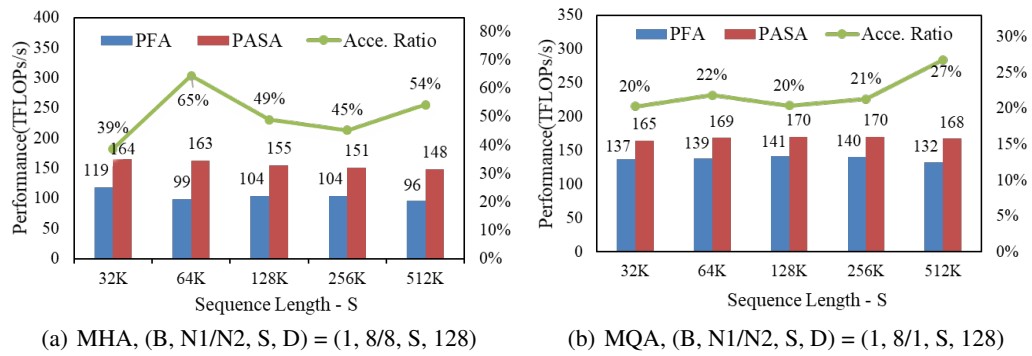

(a) MHA, (B, N1/N2, S, D) = (1, 8/8, S, 128)      (b) MQA, (B, N1/N2, S, D) = (1, 8/1, S, 128)

Figure 9: Performance of PASA and CANN-PFA(High-precision Mode).
.

developed by Ascend C(Ascend, 2024) on Ascend NPU. We evaluated both multi-head attention (MHA) and multi-query attention (MQA) for generality. The input sequence lengths range from $S = 32k$ to $512k$. The performance comparison illuminated in Figure 9 demonstrates a substantial speedup for PASA in both MHA and MQA. Specially, the acceleration ratio ranges from $1.2\times$ to $1.27\times$ and $1.39\times$ to $1.65\times$ for MHA and MQA, respectively, where the definition of FLOPs is kept consistent with FA3.0(Shah et al., 2024) as $4B \times S^2 \times N1 \times D$. Correspondingly, a maximum performance of 170 TFLOPs/s on Ascend NPU is achieved. More performance results are presented in Section K. These performance improvements stem primarily from three aspects: reduced memory-bandwidth pressure, more efficient pipelining operation on vector units and higher computing power utilization. We believe that these benefits can help alleviate similar bottlenecks across most AI accelerators.

## 4 DISCUSSION, LIMITATIONS AND CONCLUSION

We put forward PASA, a Full-FP16 attention algorithm achieving mathematical equivalence to FA without incurring overflow for long-sequence inference. Our key innovations, online pseudo-average shifting and global recovery, address numerical instability caused by sequence-dimension bias, reducing memory and data movement overhead on AI accelerators. Experiments show $1.2\times$-$1.65\times$ speedup over vendor FA, with further analysis finding a `resonance` mechanism between query and key matrices that explains overflow and accuracy degradation.

While validated on LLMs and multi-modal models, the generality of `resonance` requires broader statistical study. Adaptive triggering of PASA (to handle non-ubiquitous overflow) and integration with FP8 quantization (e.g., combined with FA3.0 are promising directions for further long-sequence extreme acceleration.

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

## A    THE PROCEDURES OF FLASH ATTENTION

The implementation of FA is able to be decomposed into the following fundamental procedures:

$$\mathbf{S}_i^j = \mathbf{Q}_i \mathbf{K}_j^T \in \mathbb{R}^{B \times N \times s_1 \times s_2} \tag{12}$$

$$\mathbf{S}_i^j = \mathbf{S}_{i,j}/\alpha \in \mathbb{R}^{B \times N \times s_1 \times s_2} \tag{13}$$

$$\mathbf{P}_i^j, l_j, m_j = softmax(\mathbf{S}_i^j) \tag{14}$$

where

$$m_j = max(m_{j-1}, rowmax(\mathbf{S}_i^j)) \in \mathbb{R}^{B \times N \times s_1 \times 1} \tag{15}$$

$$\mathbf{P}_i^j = exp(\mathbf{S}_i^j - m_j) \in \mathbb{R}^{B \times N \times s_1 \times s_2} \tag{16}$$

$$l_j = exp(m_j - m_{j-1})l_{j-1} + rowsum(\mathbf{P}_i^j) \in \mathbb{R}^{B \times N \times s_1 \times 1} \tag{17}$$

Correspondingly, the temporary output can be obtained as:

$$\mathbf{O}_i = exp(m_j - m_{j-1})\mathbf{O}_i + \mathbf{P}_i^j \mathbf{V}_j \in \mathbb{R}^{B \times N \times s_1 \times d} \tag{18}$$

Finally, after sweeping the whole row of the attention matrix, the consequent attention output can be calculated by

$$\mathbf{O}_i = \mathbf{O}_i/l_{N_{KV}} \in \mathbb{R}^{B \times N \times s_1 \times d} \tag{19}$$

Where $B \times N \times s_1 \times d$ and $B \times N \times s_2 \times d$ are the shapes of the basic block for $\mathbf{Q}$ and $\mathbf{KV}$, respectively. Consequently, the total numbers of basic blocks are $N_Q = S_1/s_1$ and $N_{KV} = S_2/s_2$. The above variables like $\mathbf{Q}_i, (\mathbf{KV})_i$ denote the basic block for the original input global matrix. The fundamental idea of FA is to fully utilize the block algorithm to make the calculation in an online method. It enhances the data locality to save memory and improves the pipelining efficiency for the computation-movement overlapping.

## B THE DATA RANGE COMPARISON FOR DIFFERENT FLOATING-POINT DATA FORMAT

Table 2: Range and Precision for Different Data Formats.

| DATA FORMATS | PRECISION | OVERFLOW BOUNDARY |
|---|---|---|
| FP8 | $6.25 \times 10^{-2}$ | 448 |
| FP16 | $4.88 \times 10^{-4}$ | 65504 |
| BP16 | $3.906 \times 10^{-3}$ | $3.4 \times 10^{38}$ |
| FP32 | $5.96 \times 10^{-8}$ | $3.4 \times 10^{38}$ |

## C   The Principle of using Optimal Accuracy Condition in PASA to Reduce Numerical Error and Overflow

To maintain the PASA effect of avoiding overflow, the hyper-parameter $\beta$ is essentially as close as the value 1.0 which represents the full subtraction of the average bias. We know that the inverse of the shifting matrix does not exist if $\beta = 1.0$. The nonexistence of the inverse will lead to the failure to obtain the correction terms. Correspondingly, the values very close to 1.0 can be selected as the $\beta$ candidates like $0.9, 0.99, 0.999$.

However, it is observed that the rounding error makes it impossible to fully represent some significant values like $(1 - \beta/s_2)$ and $(-\beta/s_2)$ in the Equation (2). The equivalent effect is that the actually used $\beta$ is changed. Nevertheless, in the correction phase of PASA, the original $\beta$ is still applied to form the correction terms.

$$f(\beta) = \frac{bn}{a(a - bn)} + \frac{1 - a}{a} \tag{20}$$

where the $n$ represents the size of the shifting matrix $\mathbf{M}$ in the Equation (2), so $n = s_2$. The parameters, $a$ and $b$, given in the Equation (20), are obtained by rounding off the original parameters in the Equation (2).

$$b = fl_{tp}(\beta/n), a = fl_{tp}(1 - \beta/n) + b, \tag{21}$$

where $fl_{tp}(\cdot)$ represents the rounding operation in low precision $tp$ determined by the data format of input $\mathbf{Q}, \mathbf{K}, \mathbf{V}$. It means that $tp = FP16$ if the data format of the input is $FP16$. In the meanwhile, $tp = BF16$ if the data format of the input is $BF16$. The whole nonlinear equation in the Equation (20) can be solved by utilizing fixed-point iteration method in the Equation (22) under high precision $cp = FP64$. We define the $Inva = \frac{\beta}{1-\beta}$ and $Inva_1 = \frac{bn}{a(a-bn)} + \frac{1-a}{a}$ as the ideal invariance and the practical invariance under rounding effect, respectively. From the Table 3, the minor numerical error is found in the practical low-precision computing of the Invariance quantity compared to the ideal quantity for the initially given hyper-parameter $\beta$. It is worth noting that the supplemented values $1 - 2^{-4} = 0.9375, 2^{-5} = 0.96875, 2^{-6} = 0.984375$ are specially chosen because they can completely denoted by the FP16 format without any numerical loss. The minor error of the invariance will cause the aliasing error in the maximum value comparison in the Equation (15), which is the dominant error source to the final output of FA. Nevertheless, if $\beta$ is solved by using Equation (22) with the initial values, the results given in the **??** illuminate that the invariance error is reduced to zero. The next part will show the end-to-end influence of the optimized hyper-parameter $\beta$. According to the above analysis, it is observed that the parameter $\beta = 0.9375$ is particularly good. It can be fully represented by FP16, and the invariance is an integer. These confirm that no rounding error is caused by the invariance itself in the correction steps of PASA.

$$\beta_{k+1} = \frac{f(\beta_k)}{1 + f(\beta_k)} \tag{22}$$

Table 3: Invariance Parameters under Initial and Optimized $\beta$ for the Computing Precision as $FP16(Inva = \frac{\beta}{1-\beta}, Inva_1 = \frac{bn}{a(a-bn)}, Rel.Err = \frac{|Inva-Inva_1|}{|Inva|})$.

| **INITIAL** $\beta$ | $Inva$ | $Inva_1$ | $Rel.Err$ | **OPTIMIZED** $\beta$ | $Inva$ | $Inva_1$ | $Rel.Err$ |
|---|---|---|---|---|---|---|---|
| 0.9 | 9.000 | 8.971 | 0.32% | 0.9 | 8.971 | 8.971 | 0.0% |
| $1 - 2^{-4}$ | 15.00 | 15.00 | 0.0% | 0.9375 | 15.00 | 15.00 | 0.0% |
| $1 - 2^{-5}$ | 31.00 | 31.25 | 0.81% | 0.96899 | 31.25 | 31.25 | 0.0% |
| $1 - 2^{-6}$ | 63.00 | 63.50 | 0.79% | 0.984497 | 63.50 | 63.50 | 0.0% |
| 0.99 | 99.00 | 102.2 | 3.23% | 0.990311 | 102.2 | 102.2 | 0.0% |
| 0.999 | 999.0 | 1031 | 3.20% | 0.999031 | 1031 | 1031 | 0.0% |

# D THE PRINCIPLE OF USING NONLINEAR EQUATION TO OBTAIN THE OPTIMAL ACCURACY CONDITION.

The numerical error could be caused by the rounding operation of shifting matrix - $\mathbf{M}$. As is mentioned above, the exact parameter, $\beta$, and the inverse of the rounded shifting matrix are simultaneously utilized in the correction procedure. If the rounding error is taken into consideration, the inverse of the shifting matrix is not like the Theorem 1. Correspondingly, the following form also holds for the rounded shifting matrix.

$$\mathbf{M}_{fp} = a\mathbf{I} - b\mathbf{J} \tag{23}$$

where $b = fl(-\mathbf{M}[0,1])$ and $a = fl(\mathbf{M}[0,0]) + b$, and $fl(\cdot)$ denotes the rounding operation using FP16. The general inverse form of the above Equation (23) can be written as the Equation (24).

$$\mathbf{M}'_{fp} = \frac{1}{a}\mathbf{I} - \frac{b}{a(a-bs)}\mathbf{J} \tag{24}$$

With the help of the Equation (6), it is natural to recover the pseudo-average values, $\frac{\beta}{1-\beta}\bar{\mathbf{S}}'^{j}_{i}$, theoretically. Nevertheless, the real pseudo-average values taking the rounding effect into account are $(\frac{bs}{a(a-bs)} + \frac{1-a}{a})\bar{\mathbf{S}}'^{j}_{i}$. To make the influence of rounding error to the minimum, the theoretical and practical pseudo average should be as close as possible. It leads to the optimization problem as

$$\underset{0\leq\beta,\beta\neq 1}{\arg\min} \quad ||f(\beta)\bar{\mathbf{S}}'^{j}_{i} - \frac{\beta}{1-\beta}\bar{\mathbf{S}}'^{j}_{i}|| = \underset{0\leq\beta,\beta\neq 1}{\arg\min} \quad |f(\beta) - \frac{\beta}{1-\beta}| \tag{25}$$

where $f(\beta) = \frac{bs}{a(a-bs)} + \frac{1-a}{a}$. The optimization problem can be solved using the iterative method in Equation (22) with a given initial guess for $\beta$. The suggested parameter, $\beta$, is as close to 1 as possible. In this situation, the term $\frac{1-a}{a}$ in $f(\beta)$ is quite small, so it is negligible compared to the dominant term.

# E  OPTIMAL ACCURACY CONDITION TO DETERMINE THE HYPER-PARAMETER $\beta$.

To solve the optimal parameter in the nonlinear Equation (20), the command `python optimal_para.py` is run for the below presented `Python` code.

```python
# optimal_para.py
import numpy as np
import torch
# To obtain the optimal alpha for PASA algorithm using fixed-point
    iteration.
# The Nonlinear Equation: beta / (1-beta) = f(beta), f(beta) = b*n / (a*(
    a-b*n))
def obtainInvPam(beta0, N, tp = torch.float16, cp = torch.float64):
    M0 = torch.tensor(1.0, dtype = cp) - beta0.type(cp) / N
    M1 = -beta0.type(cp) / N
    M0 = M0.type(tp)
    M1 = M1.type(tp)
    b = -M1.type(cp)
    a = M0.type(cp) + b
    Inv_Pam = b * N / (a * (a - b * N)) + (1 - a) / a
    return Inv_Pam
def optimal_beta(beta0, N, tol = 1.0e-8, tp = torch.float16, cp = torch.
    float64):
    err = 1.0
    iter = 0
    Inv_Pam = obtainInvPam(beta0, N, tp, cp)
    while (err > tol):
        Inv_Pam = obtainInvPam(beta0, N, tp, cp)
        beta = Inv_Pam / (1.0 + Inv_Pam)
        err = torch.abs(beta - beta0) / torch.abs(beta0)
        beta0 = beta * 1.0
        iter += 1
    return beta
if __name__ == "__main__":
    # float16
    print("======================float16(1,5,10)==================")
    print("Initial beta = 1-1/2**4, 1-1/2**5, 1-1/2**6")
    bits = int(3)
    beta0 = torch.zeros(bits)
    for i in range(bits):
        beta0[i] = 1.0 - 1.0 / 2**(i+4)
    N = int(128)
    M = beta0.size()
    M = M[0]
    beta = torch.zeros(M)
    for i in range(M):
        beta[i] = optimal_beta(beta0[i], N, tp = torch.float16)
    print("======================Results:=====================")
    print(f"for float16, initial beta: {beta0}")
    print(f"for float16, beta: {beta}")
```

## F  UNCOUPLED ARCHITECTURE OF ASCEND NPU

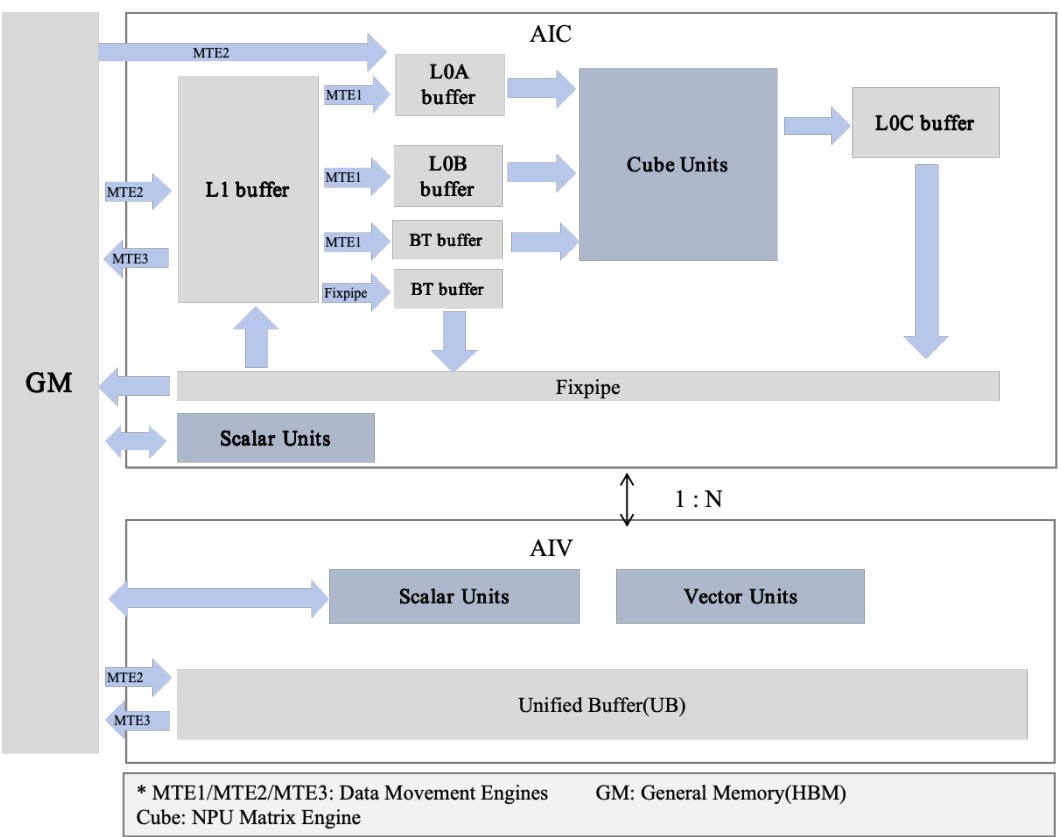

Figure 10: Uncoupled Architecture of Ascend NPUAscend (2024; 2022).

## G    NUMERICAL ACCURACY RESULTS FOR RANDOM BENCHMARK CASES

The uniform and hybrid data distribution can be found in Equation (9) and Equation (10). It is worth noting that the hybrid data distribution with the range $Am = 10$ and the mean value $x_0 = 0$ in Figure 12a is same as the random data in FA3.0Shah et al. (2024). As depicted in Figure 11a, we choose different mean values for a fixed range of $Am = 0.5$ for uniform random data distribution. The results indicate that overflow phenomenon appears when the mean value $x_0$ reaches 30 only for Partially low-precision allocation of FA(FA16-FP32). In contrast, PASA has a similar capability of avoiding overflow with original high-precision allocation of FA(FP32) widely adopted in FA on GPUDao et al. (2022b); Tri (2024); Shah et al. (2024). When it comes to the numerical accuracy for cases without overflow, it is observed that PASA has a smaller RMSE than Partially low-precision FA(FA16-FP32) for all cases with non-zero mean values, but larger than the original FA(FP32). Similar trends are also illustrated in Figure Figure 11b with increasing the range $Am$ for a relatively small mean value $x_0 = 20$. Overflow occurs when $Am$ is larger than 10 for Partially low-precision allocation of FA(FA16-FP32), while the phenomenon does not appear in other two precision allocation methods. For all cases in Figure Figure 11b, the numerical accuracy in PASA is higher than Partially low-precision allocation of FA(FA16-FP32), but is not as accurate as original high-precision allocation of FA(FP32).

For random hybrid normal-Bernoulli distribution, when the range $Am$ is fixed to a small value 10, the trends for the overflow and numerical accuracy in Figure 12a are consistent with these in Figure Figure 11a for uniform distribution. When we fix the mean value $x_0$ to 20, the overflow appears for the range $Am$ exceeding to 20 for partially low-precision allocation of FA(FP16-FP32) in Figure 12b, while PASA and original high-precision allocation of FA(FP32) still keep normal computation free from overflow or final NAN results.

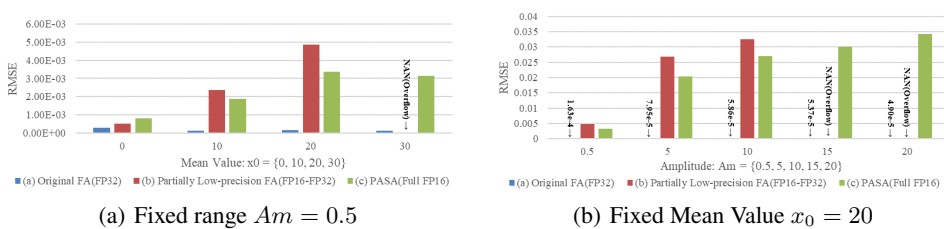

(a) Fixed range $Am = 0.5$    (b) Fixed Mean Value $x_0 = 20$

Figure 11: RMSE Comparison for Three Precision Allocations of FA Algorithms for Uniform Random Data Distribution(Left Figure: Fixed range $Am$ and Varying Mean Value $x_0$; Right Figure: Fixed Mean Value $x_0$ and Varying range $Am$. The NAN is not explicitly plotted and replaced by a text mark).

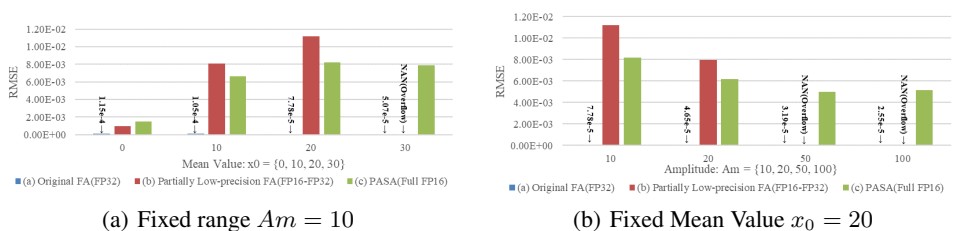

(a) Fixed range $Am = 10$    (b) Fixed Mean Value $x_0 = 20$

Figure 12: RMSE Comparison for Three Precision Allocations of FA Algorithms for Hybrid Random Data Distribution(Left Figure: Fixed range $Am$ and Varying Mean Value $x_0$; Right Figure: Fixed Mean Value $x_0$ and Varying range $Am$. The NAN is not explicitly plotted and replaced by a text mark).

# H STATISTICS FOR THE NAN APPEARANCE PERCENTAGES OF FA OUTPUT FOR RANDOM BENCHMARK DATASETS IN PARTIALLY LOW-PRECISION FA(FP16-FP32).

Table 4: NAN Percentages of FA Output for the Datasets with Uniform and Hybrid Random Distributions for Partially Low-precision FA(FP16-FP32)(Uniform: Uniform Random Distribution in Equation (9); Hybrid: Hybrid Normal-Bernoulli Random Distribution in Equation (10)).

| $N0$ | **DISTRIBUTION** | $x_0$ | $Am$ | **NAN PERCENTAGE** | **OVERFLOW**? |
|------|------------------|-------|------|--------------------|---------------|
| 1 | Uniform | 30 | 0.5 | 100% | Yes |
| 2 | Uniform | 20 | 15 | 0.12% | Yes |
| 3 | Uniform | 20 | 20 | 8.14% | Yes |
| 4 | Hybrid | 30 | 10 | 100% | Yes |
| 5 | Hybrid | 20 | 50 | 0.04% | Yes |
| 6 | Hybrid | 20 | 100 | 1.11% | Yes |

# I CLOUD MAPS FOR REAL LMS(QWEN2 AND STABLE-VIDEO-DIFFUSIO-IMG2VID MODELS).

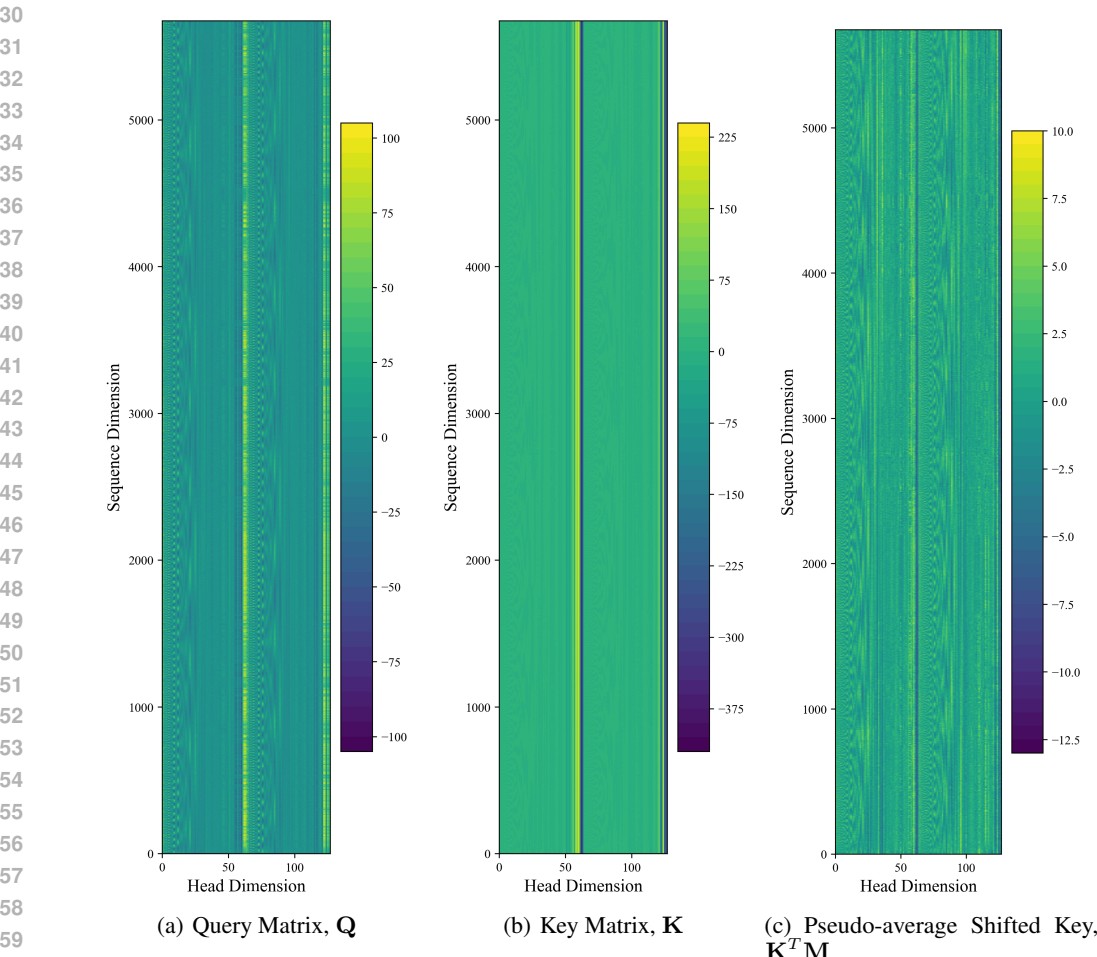

(a) Query Matrix, **Q**

(b) Key Matrix, **K**

(c) Pseudo-average Shifted Key, $\mathbf{K}^T\mathbf{M}$

Figure 13: The Data Distribution of Query and Key Matrices for Qwen2 Overflow Cases($Batch = 0$, $Head = 10$, hyper-parameter - $\beta = 0.984497$).

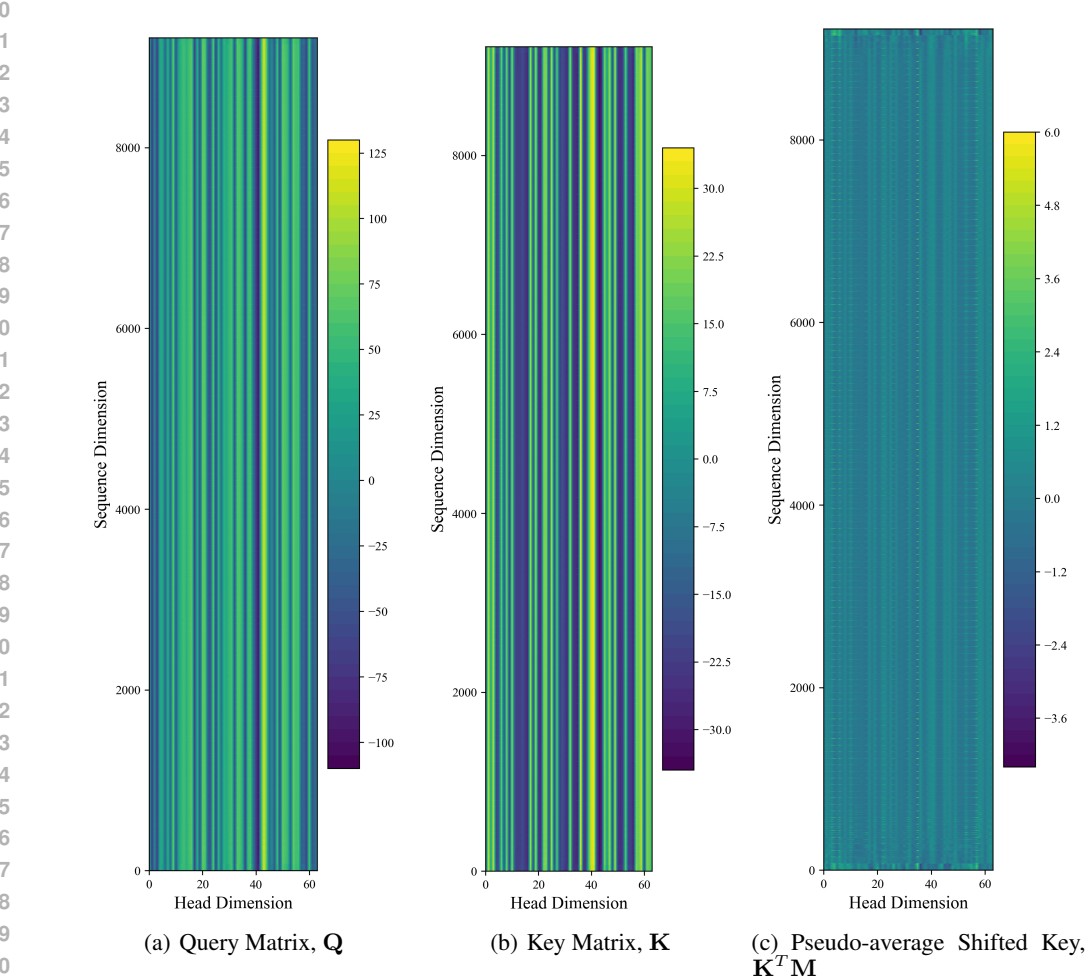

(a) Query Matrix, $\mathbf{Q}$

(b) Key Matrix, $\mathbf{K}$

(c) Pseudo-average Shifted Key, $\mathbf{K}^T\mathbf{M}$

Figure 14: The Data Distribution of Query and Key Matrices for SVD-IMG2VID Overflow Cases($Batch = 1$, $Head = 4$, hyper-parameter - $\beta = 0.984497$).

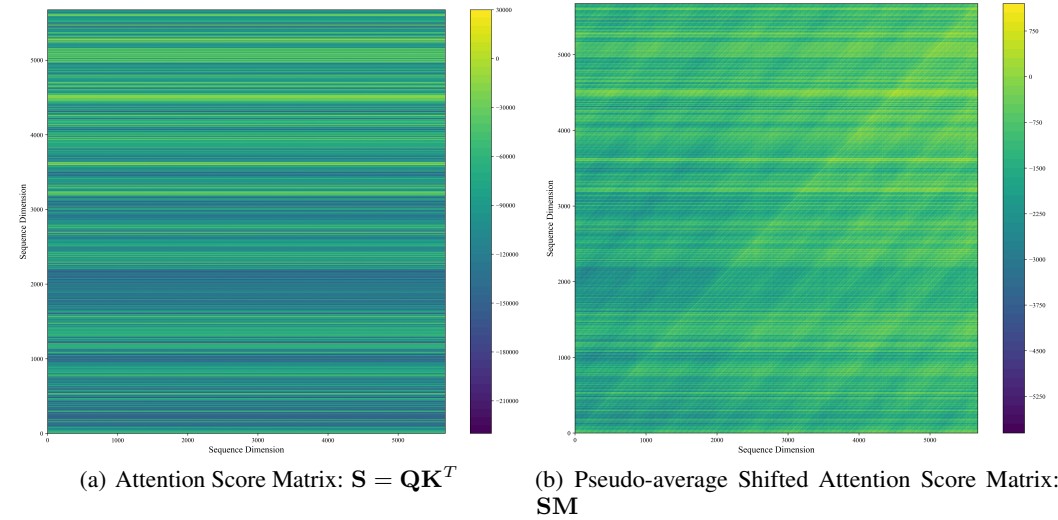

(a) Attention Score Matrix: $\mathbf{S} = \mathbf{Q}\mathbf{K}^T$

(b) Pseudo-average Shifted Attention Score Matrix: $\mathbf{SM}$

Figure 15: The Data Distribution of Original and Preprocessed Attention score matrices for Qwen2 Overflow Cases($Batch = 0$, $Head = 10$, hyper-parameter - $\beta = 0.984497$).

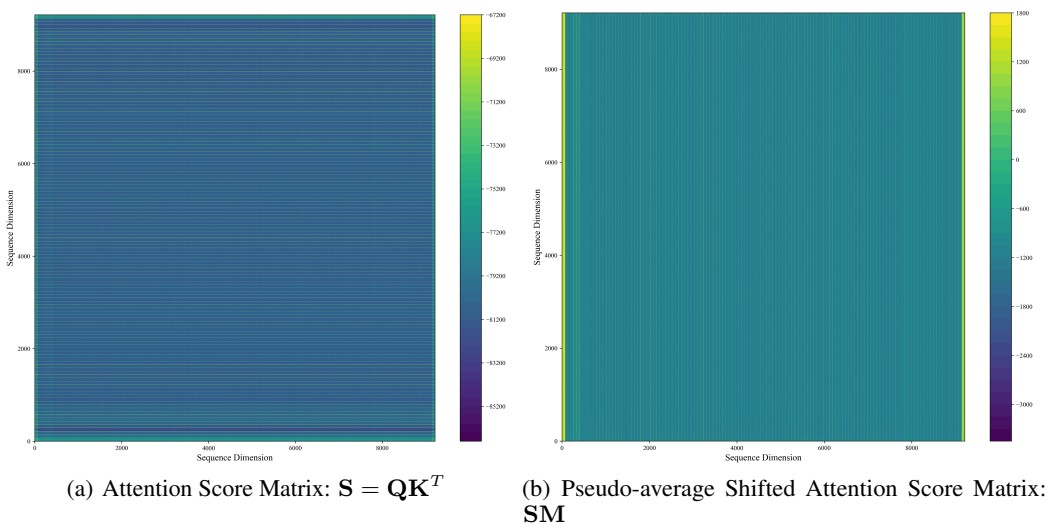

(a) Attention Score Matrix: $\mathbf{S} = \mathbf{Q}\mathbf{K}^T$      (b) Pseudo-average Shifted Attention Score Matrix: **SM**

Figure 16: The Data Distribution of Original and Preprocessed Attention score matrices for SVD-IMG2VID Overflow Cases($Batch = 1$, $Head = 4$, hyper-parameter - $\beta = 0.984497$).

## J    ORIGINAL AND PROCESSED DATA DISTRIBUTION ACROSS HEADS AND SEQUENCE POSITION IN OVERFLOW CASES FOR QWEN2-7B MODEL

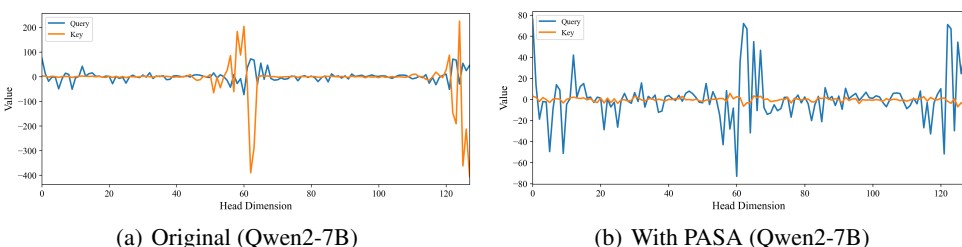

(a) Original (Qwen2-7B)                    (b) With PASA (Qwen2-7B)

Figure 17: Sampling data distributions for Query and Key across heads and sequence positions in overflow cases for Qwen2-7B Case.

# K PERFORMANCE RESULTS FOR MORE TENSOR SHAPES

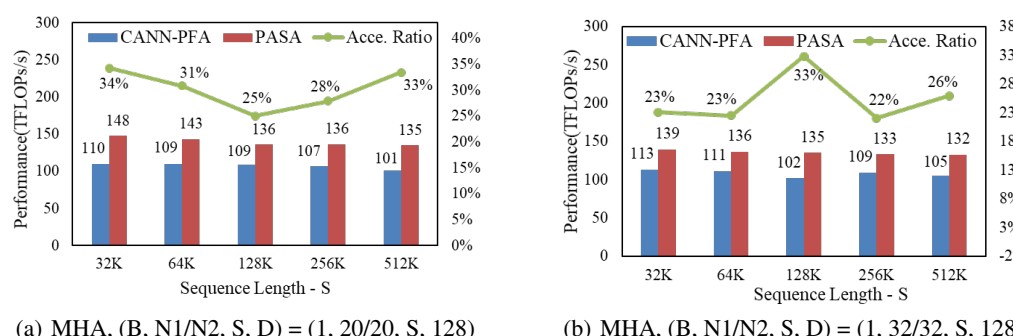

(a) MHA, (B, N1/N2, S, D) = (1, 20/20, S, 128)    (b) MHA, (B, N1/N2, S, D) = (1, 32/32, S, 128)

Figure 18: Performance of PASA and CANN-PFA(High-precision Mode).

## L  EXPERIMENTAL PARAMETER SETUP FOR THE LM INFERENCE CASES(QWEN2-7B AND SVD-IMG2VID).

The prompt from the LongBenchBai et al. (2023b) dataset is applied to validate the inference accuracy with PASA for Qwen2-7B language model. It is found that almost all the test cases from the LongBenchBai et al. (2023b) will lead to the occurrence of overflow for partially low-precision or naively full precision allocation of FA. We only present the representative one with more than $5k$ sequence length. The simplified prompt information is:

**[Prompt]:** Answer the question based on the given passage.  Only give me the answer and do not output any other words.  The following are some examples. {Passage:  Adam's apple The laryngeal prominence (commonly referred to as Adam's apple), a feature of the human  ...  ...    The visitor center is open daily (except Thanksgiving Day, December 25, and January 1) from 9:00 a.m.  to 5:00 p.m, with extended hours between Memorial Day and September 30.  During the summer season the visitor center is open until the laser light show, One River, Many Voices, ends. Show times vary, learn more >> Question:  In which country is the Grand Coulee Dam Answer:}

Both the original output from Qwen2-7B inference with high-precision attention and the output with PASA in FP16 precision are: United States. It indicates that the PASA has the same inference accuracy with the original high-precision attention for language model.

When it comes to the SVD-IMG2VID test cases, the benchmark input is given in the picture form. The official python script in Huggingface for the inference is given as:

```python
import torch
from diffusers import StableVideoDiffusionPipeline
from diffusers.utils import load_image, export_to_video

pipeline = StableVideoDiffusionPipeline.from_pretrained(
    "stabilityai/stable-video-diffusion-img2vid", torch_dtype=torch.
    float16, variant="fp16"
)
pipeline.enable_model_cpu_offload()
pipeline.unet.enable_forward_chunking()

image = load_image("https://huggingface.co/datasets/huggingface/
    documentation-images/resolve/main/diffusers/svd/rocket.png")
image = image.resize((1024, 576))

generator = torch.manual_seed(42)
frames = pipeline(image, decode_chunk_size=1024, generator=generator,
    num_frames=25).frames[0]

export_to_video(frames, "video.mp4", fps=7)
```

## M    SYSTEM AND LIBRARIES

We conducted the experiments on Atlas 800I A2 server with Ascend NPU(Uncoupled Architecture). We also utilized the latest released software stack and libraries. More specifically, they include:

    a. Hardware: Atlas 800I A2 (4 $\times$ Kunpeng CPU + 8 $\times$ Ascend NPU).

    b. Hardware Performance Per. NPU: 354TFLOPs/s(FP16, Cube) + 22TFLOPs/s(FP16, Vector).

    c. Main Memory: HBM = 64GB

    d. Framework: Torch-NPU 2.1.0 and PyTorch 2.1.0.

    e. Ascend Software Stack: CANN 8.0.0(Ascend C).

