# OpenReview forum: "Online Pseudo-average Shifting Attention(PASA) for Robust Low-precision LLM Inference: Algorithms, Numerical Analysis and Performance"
_ICLR.cc/2026/Conference — Submitted to ICLR 2026_

### Official Review · Reviewer_3xtq · 2025-10-23

**Soundness:** 3
**Presentation:** 3
**Contribution:** 3
**Rating:** 6
**Confidence:** 3

**Summary:**

This paper addresses the numerical overflow that prevents attention mechanisms like FlashAttention from being executed entirely in a low-precision (FP16) pipeline. The work identifies the root cause as a “resonance” phenomenon: a phase alignment between Query and Key vectors along the head dimension, which catastrophically amplifies their inner product. The proposed method, PASA, introduces an online pseudo-average shifting mechanism, implemented as a single hardware-friendly GEMM on the Key matrix to preemptively suppress the magnitude of attention scores. This is coupled with a novel global recovery scheme that mathematically corrects these local shifts during the online tiling process, thereby ensuring correctness. By enabling stable, full-FP16 computation, PASA eliminates the memory bandwidth bottleneck inherent to mixed-precision approaches and demonstrates significant speedups of 1.2x–1.65x over highly optimized vendor libraries on Ascend NPUs.

**Strengths:**

1. The paper’s primary strength lies in its novel and mechanistic diagnosis of the overflow problem in low-precision attention. By defining the issue as “resonance,” which refers to a specific phase alignment between Query and Key vectors, the authors provide a more concrete and actionable insight than the generic observation that values can become large. This diagnosis is convincingly supported by empirical visualizations from real-world models.

2. The proposed solution, PASA, is a well-engineered response to this finding. Its core technical contribution is twofold: a): it cleverly reframes the corrective bias subtraction as a single, hardware-efficient GEMM operation, showcasing strong algorithm and hardware co-design. b): This is correctly paired with a non-trivial online recovery mechanism that ensures mathematical fidelity is maintained within the constraints of a tiled computing framework.

3. The significance of this work is therefore highly practical, as it presents a robust method to enable fully FP16 attention. This directly addresses a critical performance bottleneck in modern AI systems by alleviating memory bandwidth pressure and allowing for better utilization of low-precision hardware, making it a valuable contribution to efficient long-sequence inference.

**Weaknesses:**

1. The empirical validation is conducted exclusively on Ascend NPUs. To substantiate the method's claimed generalizability, its performance and stability should also be benchmarked on other widely used hardware, such as NVIDIA/AMD GPUs and CPUs. This is critical because the practical speedup from a full-FP16 pipeline is highly dependent on specific architectural characteristics, and demonstrating efficacy on market-leading platforms would significantly broaden the paper's impact.

2. The primary motivation, the "resonance" phenomenon, is illustrated using sampled 1D plots (e.g., Figure 7) that likely represent worst-case scenarios rather than the overall data distribution. While illustrative, this selective sampling may overstate the issue's pervasiveness. Providing more comprehensive visualizations, such as heatmaps of the attention score matrix or distributions of max scores across all tokens, would offer more robust evidence that resonance is a systematic, rather than an anecdotal, problem.

3. The paper’s scope is confined to the FP16 data type, while many modern systems and training setups increasingly favor BF16 due to its larger dynamic range. The authors do not discuss whether the resonance issue persists in BF16 (e.g., as a source of precision loss instead of overflow) or if the PASA mechanism would still be beneficial. A discussion of the method's relevance and potential adaptations for BF16 would provide a more complete picture of its utility.

4. While the chosen models (Qwen2-7B, SVD) are suitable for demonstrating the core problem, the validation could be strengthened by including a wider range of modern model architectures. Testing PASA on structurally different models, such as those employing Mixture-of-Experts (MoE) or alternative normalization schemes, would provide stronger evidence for the robustness and general applicability of the proposed technique beyond the specific cases presented.

5. The performance claims are made against a vendor library, but a comparison with the open-source state-of-the-art, FlashAttention-3, is conspicuously absent. Modern kernels like FlashAttention-3 achieve high performance by carefully overlapping the compute-intensive GEMM operations with the memory-bound softmax and accumulator updates. In such a highly optimized pipeline, the latency of the mixed-precision (FP32) softmax updates may already be partially hidden. Therefore, while PASA’s benefit of halving the memory traffic from these intermediates is clear, the real-world speedup on top of FlashAttention-3 is not guaranteed and must be empirically demonstrated to be significant.

6. There are no comparative results at the whole-network level, such as using the Qwen2-7B model on commonly used datasets and comparing with FP32 Softmax, e.g., MMLU, HumanEval, GSM8K, Multi-Exam, etc. (see [Qwen2-7B on Huggingface](https://huggingface.co/Qwen/Qwen2-7B) for reference).

**Questions:**

1. Given that Q is in the outer loop of the tiled algorithm (FlashAttention), applying a single corrective operation to it seems, in theory, more efficient than modifying blocks of K in the inner loop. A discussion on this trade-off, and perhaps an ablation study in an inference/training setting, would significantly strengthen the justification for your approach.

2. PASA introduces new computations, namely the K * M multiplication and the online correction term calculations. Please quantify the performance overhead of these two operations. A breakdown of their latency, and an analysis of whether they are successfully hidden/overlapped within the main attention pipeline, is necessary to fully assess the net performance gain.

3. The core argument for the performance gain is the reduction in memory bandwidth from eliminating FP32 intermediates. To help quantify this claim more precisely, could you provide:

   - A micro-benchmark showing the latency difference between the softmax-related updates (i.e., finding max, calculating exp, and sum) when the accumulators and intermediate score matrix are in FP16 versus FP32?
   - An approximate profiling breakdown of the baseline mixed-precision attention kernel, showing the percentage of time spent on GEMM computation versus the non-GEMM parts (memory movement, vector math for softmax, etc.)? This data would more directly substantiate the claim that the latter is indeed a significant bottleneck that PASA effectively addresses.

4. Could you provide comparative results for Qwen2-7B on standard datasets such as MMLU, HumanEval, GSM8K, and Multi-Exam, in comparison with FP32 Softmax?

---

> ### Author Response · Authors · 2025-11-21
>
> **1.4.1 On Strength: Originality and Innovation**
>
> Thank you for your professional comments. PASA enables **full low-precision (FP16) attention** on NPUs (GPU validation planned) by:
>
> - Solving numerical instability/accuracy issues
> - Achieving **significant speedup vs. CANN** (Vendor Library)
> - Deriving an **optimal accuracy condition** (nonlinear eq.) for FP16
> - Discovering **resonance** as an FP16 instability mechanism
> - Combining mathematical proofs + experiments
>   → First full low-precision attention on modern AI chips
>
> **1.4.2 On Weakness**
>
> * 1. PASA is math-equivalent to standard/Flash Attention. Algorithm 1 targets hardware-independent optimizations (HBM-local transfer, pipelining). Due to page limits, we focus on algorithms and partial NPU validation; GPU porting is future work.
>
> * 2. Complete data distribution heatmaps are in **appendix I** for language and video models.
>
> * 3. PASA aims for stable full fp16 attention. BF16 showed significant accuracy loss in early tests and is unnecessary if fp16 works well.
>
> * 4. Current work establishes "resonance" existence; systematic experiments with latest models and GPU porting are planned.
>
> * 5. Modern chips (NPU/GPU) enhance matrix power while low-bit quantization reduces matmul overhead, making attention vector/memory-bound - PASA addresses this bottleneck.
>
> * 6. See **Figure 8** (SVD) and **appendix L** (Qwen2-7B) for network-level results. More E2E benchmarks are future work.
>
> **1.4.3 On Questions**
>
> * 1. Modifying **K** (not Q) suffices due to softmax shifting invariance (**Equation 1**).
>
> * 2. **Overhead**: K-matrix shifting overhead is negligible for long sequences:
>
>   | seq_length | Overhead |
>   | :--------: | :------: |
>   |    16K     |    5%    |
>   |    32K     |    4%    |
>   |    64K     |    3%    |
>   |    128K    |    2%    |
>
> * 3. (1) The micro-benchmark cannot be provided due to the strong coupling of the pipelining operations for current stage. Instead, we can obtain the contribution of fp16 vector computation to the total performance gain by subtracting the overhead evaluated as above table("**Overhead**") from the overal performance improvement given in **Figure 9**. Hence, the performance gain contributed from replacing fp32 with fp16 is about 15%~60%. Particularly, from the overhead statistics, we can see that PASA may cause the performance deterioration for short sequence inference. More exacted, "Long sequence" = $32K \sim 512K$ has been given in our paper. Shorter sequences ($4K-32K$) show: (1) Speedup ↓ with sequence length; (2) Overhead negligible for long sequences
>
>   | seq_length | Speedup(t1/t2 - 1) | Times(us), t2, PASA(fp16) | Times(us), t1, CANN-PFA |
>   | :--------: | :----------------: | :-----------------------: | :---------------------: |
>   |     4K     |      4.0248%       |         389.7899          |        405.4782         |
>   |     8K     |      14.1759%      |         1108.9602         |        1266.1652        |
>   |    16K     |      15.5488%      |         3854.8809         |        4454.269         |
>   |    32K     |      30.4024%      |        14003.0098         |       18260.2648        |
>
> ​	(2) The attention is complicated for the pipelining of matrix and vector cores. During developing PASA on real NPU silicon, we mainly utilized the NPU tracing simulation tool to optimize the six pipelines as shown in **Figure 10**. Here we picked the data from case with sqeuence length equal to 64K to illuminate the optimization of PASA on original FA, due to intermediate data(attention matrix score) type changing from FP32 to FP16, the fixp time decreased by 60%, which further helps reduce the mac time for matrix unit(CUBE). By contrast, the vector time is only reduced by 9%, which is due to the increased operations of PASA for calculating the global correction terms. Last but not least, it is worth noting that full low precison is much easier for the pipelining of different operations on NPU. When it comes to GPU(future work), the optimization is perhaps also suitable.
>
> |                     | aic_mac_time(us) | aic_fixp_time(us) | aic_vec_time(us) |
> | :-----------------: | :--------------: | :---------------: | :--------------: |
> | CANN-PFA(FP16-Fp32) |    32594.365     |     55729.93      |    44311.978     |
> |     PASA(FP16)      |    26125.313     |     34748.259     |    40704.507     |
> |    Acceleration     |      1.25x       |       1.60x       |      1.09x       |
>
> * 4. We have tested the **LongBench(2023)** datasets designed for long-sequence text inference for Qwen2-7B compared to the fp32 E2E results. The comparison can be found in **Appendix L**. From our current experiments and evidence, we found that the "resonance" induced numerical stability for low precision computation occurs more frequently in video models than language models. Hence, we do not test too much text datasets for Qwen2-7B.

---

> > ### Comment · Reviewer_3xtq · 2025-11-25
> >
> > Thank you for the authors’ response, which has addressed some of my concerns. However, I will maintain my score for the following reasons:
> >
> > ---
> >
> > > PASA aims for stable full fp16 attention. BF16 showed significant accuracy loss in early tests and is unnecessary if fp16 works well.
> >
> > The authors have not fully addressed my concerns. Beyond BF16, there are other low-precision formats, yet the method is only discussed in the context of FP16. This raises concerns about whether the approach is specifically designed for FP16, which appears inconsistent with the claim that “PASA is math-equivalent to standard/Flash Attention.”
> >
> > > Current work establishes "resonance" existence; systematic experiments with latest models and GPU porting are planned.
> >
> > In the past year, MoE (Mixture-of-Experts) architectures have been predominantly used in large-scale industrial models. The manuscript lacks experiments on MoE models, so the effectiveness of the proposed method in this context remains unclear.

---

> > > ### Author Response · Authors · 2025-11-26
> > >
> > > > 1. "The authors have not fully addressed my concerns. Beyond BF16, there are other low-precision formats, yet the method is only discussed in the context of FP16. This raises concerns about whether the approach is specifically designed for FP16, which appears inconsistent with the claim that “PASA is math-equivalent to standard/Flash Attention."
> > >
> > > For the concerns about the FP16 precision, we believe that the current PASA work is quite straightforward to be extended to support more precision investigation like FP8 or FP8-FP16 hybrid manner in the full attention procedures. However, it needs more statistical studies for inference accuracy, because it seems too radical considering that the precision allocation of SOTA attention like FA2.0/FA3.0 is still dominated by FP32 not FP16 inside the kernel. The PASA work is trying to push the highest precision to FP16, which is definitely a big jump for attention acceleration on modern hardwares. When it comes to other precisions(maybe new format), we believe the most meaningful thing is to integrate FP16 PASA with (QK) quantization precision like INT8/FP8/FP6/FP4 to form the mixed-precision attention whose highest precision is FP16 not the current FP32. PASA is the method to provide the good routine.
> > >
> > > > 2. "In the past year, MoE (Mixture-of-Experts) architectures have been predominantly used in large-scale industrial models. The manuscript lacks experiments on MoE models, so the effectiveness of the proposed method in this context remains unclear."
> > >
> > > When it comes to MoE model, it is a pity that we have not covered this widely-adopted method in language models like Deepseek and ChatGPT. Nevertheless, it is worth noting that we did not find the numerical overflow for FP16 in other language models including other Qwen and Llama models. For stable diffusion model(video model), we have tested SVD, open-sora as well as Tencent-Hunyuan, and found that low-precision(FP16) overflow occurs in these models due to "resonance" mechanism between Query and Key tensors. Hence, we conclude that the "resonance" more likely appears in video generation models not language models. The mechanism to answer this difference is to be done in the future.
> > >
> > > Last but not least, thanks for your professional comments for PASA especially for the low-precision algorithm and the finding of "resonance" phenomenon.

---

### Official Review · Reviewer_m9Dt · 2025-10-29

**Soundness:** 3
**Presentation:** 3
**Contribution:** 1
**Rating:** 2
**Confidence:** 3

**Summary:**

PASA is an algorithm for computing attention with low-precision data types like FP16, while benefitting from the high FLOPs throughput of algorithms such as FlashAttention. PASA builds on prior work SageAttention, that increases numerical stability by shifting key vectors by a constant vector and reducing the chance of overflows in the attention score matrix. PASA extends SageAttention by shifting key vectors by an online block-wise average of the key vectors, promising higher stability. PASA is evaluated on a synthetic QKV dataset, and Ascend NPUs with Qwen2-7B and IMG2VID models.

**Strengths:**

Thank you for submitting your work. I found the in-depth technical explanation to be clear, and the idea of calculating averages online sounds promising.

**Weaknesses:**

However, I have three key questions:
* I didn't see performance or accuracy comparisons to prior work, including SageAttention. Have I missed anything? Since the ideas are similar (mean shifting), it would be important to show that the **online** aspect of PASA is worth the extra computations.
* Why is the evaluation limited to Ascend NPUs? Prior work, including SageAttention, focus on NVIDIA GPUs which have had wider support for various kernels and serve as more stable and more optimized baselines.
* I observed that the evaluation and technical discussion is focused on FP16. Recent LLMs are commonly served in BF16 format. It would be critical to show that the numerical instability problem still exists with BF16 and PASA can reduce it.

Some minor comments:
* I would strongly suggest proofreading the paper for grammar. While it was possible to understand the technical content, the presentation and flow would benefit from it.
* Typo in equation 3

**Questions:**

Please see weaknesses.

---

> ### Author Response · Authors · 2025-11-16
>
> We sincerely appreciate for your valuable comments. We have addressed key clarifications and outline<u>d</u> the revisions we will make to enhance the clarity and completeness of the paper. We will also correct the typos, writing and typesetting issues in the final version.
>
> **1.3.1 On Strength: Originality and Innovation**
>
> Thank you for the feedback. PASA enables **full low-precision (FP16) attention** on NPUs by:
>
> 1. Solving numerical instability/accuracy issues
> 2. Achieving **significant speedup vs. CANN**(Vendor Library)
> 3. Deriving an **optimal accuracy condition** (nonlinear eq.) for FP16
> 4. Discovering **resonance** as an FP16 instability mechanism
> 5. Combining mathematical proofs + experiments
>    → First full low-precision attention on modern AI chips
>
> **1.3.2 On Weaknesses**
> * 1. SageAttention (FP8) as well as other int8/int4 quantization methods on Q, K are orthogonal to PASA – combinable but not our focus for current stages. In our current paper, the performance comparison is mainly conducted with the flash attention implementation on NPU without any quantization like SageAttention or low-precision like PASA. We have mentioned that we will do the integration with quantization on Q, K like SageAttention in our future work in our submitted paper.
> * 2. PASA is math-equivalent to standard/Flash Attention (hardware-independent). Algorithm 1 targets optimizations shared by NPU/GPU:
>
>     - HBM-local memory transfer
>     - Pipelining (e.g., double buffering)
>
>     Hardware-specific tuning (tiling/programming models) differs across SIMD (NPU) and SIMT (GPU). GPU porting is future work. This work advances robust low-precision inference infrastructure. We welcome deeper engagement with PASA's math and extension to more AI chip architectures.
>
> * 3. The numerical stability will not occur if BF16 is adopted, but it will incur very large rounding error if the full procedures of the attention calculation(we have conducted similar experimental in the very early development stage of PASA) due to the sentivity of $QK^T$/softmax rounding. The *resonance* is not strongly related to numerical format, and it can lead to the numerical stability(overflow) for low precision format.
>
> Some minor comments:
>
> * We will do the proofreading in our final submitted paper. Thanks for your suggestion.
>
> * Typo in equation 3 will be revised for the $\beta$.

---

> > ### Comment · Reviewer_m9Dt · 2025-11-17
> >
> > Thank you for your response. Unfortunately, they do not address my concerns. Please see my comments below.
> > 1. **SageAttention is not orthogonal to PASA**. SageAttention proposes the idea of mean-shifting, and PASA builds on SageAttention by applying **online** mean-shifting. Without a comparison to SageAttention, it's hard to argue that PASA's core contribution, online mean-shifting, is really needed.
> > 2. PASA may be math equivalent to FlashAttention, but still slower on an NVIDIA H100 GPU. While I understand that some tuning is required, it is a necessity to show that PASA's edge over FlashAttention is not a byproduct of a poorly optimized FlashAttention backend for Huawei GPUs. Please note that prior work (FlashAttention, SageAttention, ...) also prove the value of their work on mainstream hardware.
> > 3. Thank you for the response. These comments, **along with supporting experiments**, should be added to the paper.

---

> > > ### Author Response · Authors · 2025-11-18
> > > **The main contributions of PASA and its relationship with SageAttention**
> > >
> > > Thank you for your time and for considering our work. We would like to provide some clarification regarding the relationship between SageAttention and our proposed method, PASA.
> > >
> > > First and foremost, PASA is an algorithm built upon the foundation of FlashAttention. Its core contribution is a pseudo-mean shifting​ technique applied to the $K$ matrix. This operation significantly reduces the risk of overflow in the $QK^T$ output when using half-precision formats, thereby enabling the entire softmax computation to be conducted in low precision. This is highly beneficial for hardware with less powerful vector computation units and aligns well with the prevailing trend of low-precision inference for large models.
> > >
> > > However, implementing "mean shifting" on AI chips is non-trivial. Given the limited size of the L1 level cache, this operation often necessitates loading the $K$ matrix from global memory to the L1 level cache twice, simultaneously increasing the computational load on the vector computation units (CUDA core in GPUs or Vector Cores in NPUs). Even for state-of-the-art AI chips like NVIDIA GPUs, vector computation can become a bottleneck in low-precision FlashAttention scenarios (e.g., 4-bit FlashAttention, Fig 4 in [Inside NVIDIA Blackwell Ultra: The Chip Powering the AI Factory Era](https://developer.nvidia.com/blog/inside-nvidia-blackwell-ultra-the-chip-powering-the-ai-factory-era/#accelerated_softmax_in_the_attention_layer)). Recognizing this, subsequent GPU architectures have introduced strengthened vector operation instructions specifically for attention mechanisms (Attention Instruction in B300 improve the 'exp2' instruction, and Rubin CPX will improve it further. See [Rubin CPX: built to accelerate long-context processing](https://developer.nvidia.com/blog/nvidia-rubin-cpx-accelerates-inference-performance-and-efficiency-for-1m-token-context-workloads/#rubin_cpx_built_to_accelerate_long-context_processing)). From a performance perspective, offloading operations with low parallelism to dedicated tensor computation units (Tensor Cores) is a key method for improving computational utilization—a direction further emphasized in the Blackwell architecture with enhanced Tensor Cores and the new TMEM unit.
> > >
> > > A significant contribution of PASA is the efficient matrix-based implementation of the "mean shifting" step. This approach is supported by rigorous mathematical derivation, including a formal proof for the pseudo-average parameters and an analytical solution for a matrix inverse, making it a theoretically sound and effective method. This technique allows the "mean shifting" operation to be seamlessly integrated into the FlashAttention computation process.
> > >
> > > SageAttention is undoubtedly an important and relevant work on efficient attention, which also employs mean shifting on the $K$ matrix to enhance the precision and numerical stability of low-precision FlashAttention-like operators. However, as discussed above, for even lower-precision operators, the overhead from the vectorized operations introduced by mean shifting may become non-negligible. Therefore, the tensor-core-based pseudo-average shifting scheme proposed by PASA, which is integrated into the FlashAttention workflow, can be viewed as a potential concrete implementation strategy for the "mean shifting" step in SageAttention. This could further enhance the performance of SageAttention in ultra-low-precision scenarios or on AI accelerators with less powerful vector units, which is the reason why we claim that PASA is orthogonal to Sageattention.
> > >
> > > Furthermore, we must acknowledge that, due to certain complex constraints, we currently do not have access to advanced NVIDIA GPUs for testing. We sincerely apologize for being unable to present direct performance measurements of PASA on GPU hardware. Nevertheless, based on our theoretical analysis and considering the ongoing strengthening of Tensor Core capabilities in hardware evolution, we believe that our approach holds promise for delivering performance benefits in low-precision FlashAttention scenarios on GPUs as well.
> > >
> > > In summary, PASA is a work that combines algorithmic innovation with hardware awareness. It integrates "mean shifting" elegantly into the FlashAttention process through mathematical transformation, backed by solid theoretical foundations, and is designed with practical hardware implementation details in mind to achieve tangible performance gains.
> > >
> > > Thank you once again for your valuable time and consideration.

---

### Official Review · Reviewer_2wog · 2025-11-01

**Soundness:** 3
**Presentation:** 3
**Contribution:** 3
**Rating:** 6
**Confidence:** 2

**Summary:**

This paper proposes PASA (Pseudo-Average Shifting Attention), an online variant of FlashAttention that enables stable full-FP16 attention without modifying model weights or sacrificing accuracy. The key idea is to pre-shift keys via a structured transformation matrix M effectively removing the mean bias responsible for overflow while remaining mathematically equivalent to FlashAttention. The paper also introduces an online recovery mechanism ensuring global consistency, with detailed theoretical derivations (Theorem 1) and a rounding-aware fixed-point method to select beta. Evaluations on Ascend NPUs demonstrate ~1.5x speedups and complete elimination of overflow across both synthetic stress tests and real models.

**Strengths:**

The strengths of this paper lie in both novelty and practical utility. It introduces pseudo-average shifting as a principled and lightweight mechanism for stabilizing FP16 attention without altering model semantics. This is a clear conceptual advance over prior FlashAttention variants and bias-correction methods, addressing overflow at its source through a mathematically grounded and hardware-compatible reformulation. The design is particularly elegant: it preserves the core pipeline of FlashAttention while embedding the correction into a single matrix operation, maintaining equivalence in exact arithmetic.

Beyond novelty, the paper stands out for its reproducibility. The presentation is detailed and clear, with Algorithm 1, the beta-selection routine, and theoretical derivations all well-documented. Overall, the work is easy to follow, theoretically sound, and practically impactful, offering a genuine improvement to the reliability and efficiency of low-precision attention.

**Weaknesses:**

While the method is well-motivated and the experiments are promising, the evaluation focuses mostly on FLOPs and synthetic benchmarks rather than end-to-end latency. Reporting real tokens-per-second or full inference latency on large models would make the results far more compelling. Comparisons to high-performance FA or SageAttention are also missing, leaving it unclear how much of the gain comes from the pseudo-average shift itself versus the FP16 pipeline. Finally, the resonance explanation, while interesting, remains qualitative and could benefit from more quantitative analysis or supporting metrics.

The end-to-end evaluation lacks rigor. While the hardware-level metrics and kernel performance are well analyzed, the paper doesn’t provide a clear picture of how PASA impacts total inference time, throughput, or memory footprint in full model runs. Without these measurements, it’s hard to assess the real deployment gains compared to existing optimized attention kernels.

To improve my score, I would like to see evaluations with other common self-attention techniques such as SageAttention and a more comprehensive model quality evaluation.

**Questions:**

1. Could the authors comment on how PASA will perform on different types of transformer-style networks? I'm specifically interested in the computational implications of running this on DiT blocks vs. standard MHSA blcoks. Does the pre-shifting step interact differently with their attention structure or memory patterns?

2. Figure 8 seems to be the only large-scale evaluation in this paper and just shows SSIM on one image. Can the authors please report more useful image generation metrics such as CLIP Score CLIP-IQA, and QSNR? It would be helpful to run a standard image generation evaluation here to confirm what is just briefly mentioned for end-to-end evaluation.

---

> ### Author Response · Authors · 2025-11-16
>
> **1.2.1 On Strength: Originality and Innovation**
>
> Thanks for your comments for the strength of PASA. We believen that PASA targets **inference bottlenecks**  with three innovations:
>
> 1. **First full-FP16 attention**: Achieves 25~65% speedup for long sequences (text/video generation) vs. SOTA methods (FA1.0/2.0/3.0) relying on FP32 for stability.
> 2. **Online shifting/recovery**: Ensures numerical stability, accuracy and performance in FP16 via:
>    - Math-property-driven compression of attention scores
>    - Rounding-error analysis matching FP32 accuracy
>    - Pipeline-friendly design for NPUs
> 3. **Resonance mechanism**: Explains FP16 instability through error amplification, validated in both single-modal (LLM) and multi-modal (video) models.
>
> **1.2.2 On Weakness**
>
> We have not finished the E2E performance evaluation for PASA because the current paper focuses more on the algorithms and operators. For E2E evaluation, we only finished the accuracy and numerical stability evaluations for PASA. Basically, for the contribution of PASA on E2E inference latency and throughput, it changes with different models and sequence lengths. Nevertheless, for video generation inference like the img2video case in our paper, the attention calculation is the dominant part, and is about over 80% of percentage of the E2E inference time from our profiling results. Considering the consistence with the previous FA series papers published by Tri Dao, we choose to show the operator performance gains in our paper.
>
> Besides, SageAttention (FP8) as well as other int8/int4 quantization methods on Q, K are orthogonal to PASA – combinable but not our focus for current stages. We have mentioned that we will do the integration in our future work in our submitted paper. To observe the quantitative contribution of PASA on performance, the best comparison object is the high-performance flash attention implementation on Ascend as adopted in our paper.
>
> * 1. PASA can be integrated into any transformer architectures with softmax operation. For these architectures without softmax like naive linear attention, the PASA cannot be adopted. This is because of the adoption of softmax invariance in Eq. (1). Particularly, for sparse attention with softmax operation, we are confident that PASA can be utilized. Besides, the principle has no diffference for the pre-shifting step interaction, but the practical high-performance implementation on real silicons can be a little different because of the neglection of some calculation steps like sparse attention.
>
>
> * 2. Due to page limit, we only put the SSIM for one typical image as published on Huggingface. Generated videos are identical (SSIM = 0.9847 > 0.95) according to vBench comparison. The details of vbench report is:
>
> ```
> ==================================================
> Video Comparison Report: PASA.mp4 vs SVD_Rocket-ori.mp4
> ==================================================
> • Analyzed Frames: 25
>
> Key Metrics:
>   - Structural Similarity (SSIM): 0.9847 (1.0 = identical)
>   - Peak Signal-to-Noise Ratio (PSNR): 43.78 dB (>40 dB indicates high quality match)
>   - Mean Squared Error (MSE): 2.76 (0 = identical)
>   - Different Pixels Percentage: 0.57%
>
> ✅ Conclusion: Video content is highly consistent (SSIM > 0.95)
> ==================================================
> ```
> Besides, due to the page limits, we have not finished a lot of experiments on video benchmarks. We want this paper to focus on the algorithm and the evidence of *resonance*. For more systematic experiments on different multi-modal models and video benchmarks, we will do that in our following work related to illuminating the origin of *resonance* on the basis of the first principle of transformer architectures.

---

### Official Review · Reviewer_qZup · 2025-11-01

**Soundness:** 3
**Presentation:** 3
**Contribution:** 2
**Rating:** 2
**Confidence:** 4

**Summary:**

Model
This paper proposes PASA (Pseudo-Average Shifting Attention), an online algorithm for low-precision attention computation aimed at enhancing numerical stability and accuracy for long-sequence LLM inference.
Summary:
Attention computation in large language models (LLMs) is a significant bottleneck, especially for long-context and multi-modal tasks, due to its quadratic complexity and memory demands. While FlashAttention (FA) has emerged as a leading optimization method, current implementations often rely on FP32 for intermediate variables to maintain numerical stability, which can become a performance bottleneck due to memory bandwidth limitations. Fully low-precision (e.g., FP16) FA, while promising greater performance and energy efficiency, faces high risks of overflow and underflow.
PASA addresses these issues by introducing two key innovations: (1) online pseudo-average shifting and (2) global recovery. These mechanisms dynamically adjust attention score statistics to prevent overflow and preserve numerical accuracy. The core idea is to leverage the translation invariance of softmax through online block-wise processing, performing bias correction at the block level. This design ensures compatibility with hardware pipelining architectures, reduces numerical errors through local pseudo-average shifting, and enables efficient implementation via batched matrix multiplications.
The paper identifies that numerical instability in attention largely stems from a "resonance mechanism" where phase alignment (or anti-alignment) between query and key matrices along the head dimension amplifies inner products, leading to large values and overflow. PASA effectively suppresses this resonance.
Experiments on Qwen2-7B and Stable-Video-Diffusion models, running on Ascend NPUs, demonstrate that PASA eliminates overflow issues, achieves a 1.2x-1.65x speedup over highly optimized vendor libraries with full FP16 precision, and maintains inference accuracy matching high-precision methods. The authors claim this is the first work to enable fully FP16 acceleration for long-sequence attention with guaranteed numerical stability.

**Strengths:**

1. PASA directly tackles the major challenge of numerical instability (overflow/underflow) when attempting full low-precision (FP16) attention, which is crucial for improving the efficiency of LLMs for long-sequence inference.
2. Demonstrating a 1.2x-1.65x speedup with full FP16 and reaching up to 170 TFLOPs/s on Ascend NPUs is a strong indicator of its practical utility for accelerating LLM inference.
3. The paper asserts that PASA achieves "mathematical equivalence to Flash Attention" and maintains accuracy comparable to high-precision methods, which is vital for real-world deployment.
4. The introduction of "online pseudo-average shifting" and "global recovery" specifically tailored to mitigate resonance-induced overflow is a clever and effective algorithmic innovation.
5. PASA is designed to be compatible with hardware pipelining and utilizes low-precision compute units effectively, demonstrating good co-design principles for AI accelerators.

**Weaknesses:**

1. The experimental validation is primarily conducted on Huawei Ascend NPUs. While results are generalized to "most AI accelerators," the extent of performance gains and stability on other platforms (e.g., NVIDIA GPUs) needs further explicit demonstration.
2. The optimal accuracy condition relies on selecting a hyperparameter β (close to 1.0) through a fixed-point iteration solved in FP64. While the paper provides chosen values, this step adds an extra layer of complexity to the deployment and optimization process.
3. While the concept of "resonance" is introduced, a more rigorous mathematical formalization or in-depth statistical analysis of its occurrence frequencies across diverse real-world datasets (beyond the two models tested) could strengthen this claim.
4. The paper states that "global bias correction conflicts with hardware pipelining." While PASA's block-level correction avoids this, a more detailed explanation of why global correction specifically conflicts with pipelining could be beneficial.
5. While compared to vendor-optimized FA (CANN-PFA), a more direct comparison with other low-precision techniques (e.g., INT8/INT4 quantization strategies mentioned in the background) that aim for similar efficiency gains, in terms of both stability and performance, would provide a more complete picture of PASA's competitive advantage.
6. While "online block-wise processing" is mentioned, further details on how the online nature impacts memory usage or latency for very long sequence inference, beyond just avoiding overflow, would be insightful.

**Questions:**

see Weaknesses

---

> ### Author Response · Authors · 2025-11-16
> **Rebuttal by Authors**
>
> **1.1.1 On Strength: Originality and Innovation**
>
> PASA resolves NPU bottlenecks for long-sequence tasks (e.g., text summarization, video generation) by pioneering **full-FP16 attention computation**. Unlike SOTA methods (FA1.0/2.0/3.0) that use FP32 for stability, PASA compresses attention score ranges via: (1) **Online pseudo-average shifting** (leveraging softmax invariants); (2) **Global recovery** (Theorem + finite-precision analysis); (3) **Single-step GEMM** efficiency (replacing vector ops). Nonlinearity prevents exhaustive proofs; thus, experiments (random data/LLMs/video-LMs) validate performance. Detailed math is in appendices due to space limits.
>
> **1.1.2 On Weaknesses**
>
> * 1. PASA is math-equivalent to standard/Flash Attention (hardware-independent). Algorithm 1 targets optimizations shared by NPU/GPU:
>
>     - (1) HBM-local memory transfer;
>
>     - (2) Pipelining (e.g., double buffering);
>
>     - (3) Hardware-specific tuning (tiling/programming models) differs across SIMD (NPU) and SIMT (GPU). GPU porting is future work.
>
> * 2. The optimal accuracy condition to derive the hyper-parameter-$\beta$ is not coupled into the whole inference workflow. On the other hand, it can be pre-calculated just once and constantly stored as a fixed parameter for inference. The two parameters which need to be known to determine the optimal $beta$ are head dimension $d$ and the basic tiling block. In summary, it will not increase the complexity of large model deployment.
>
> * 3. *Resonance* defined in L390 with schematic illustration. It's a physical concept (e.g., mass-spring systems) describing perfect correlations. In attention, resonance emerges from Q/K distributions, causing numerical instability through value amplification in low precision. Observed in both single-mode (language) and multi-mode (text-video) models. We also have conducted statistical experiments on open-sora and Tencent-Hunyuan video models. Similar overflow phenomenon and data distribution have been observed. Nevertheless, for language models, the *resonance* is unusual like Qwen. Hence, we have not covered all the popular large models because we believe that the more important thing is to find the real origin of *resonance* on the basis of understanding the difference of transformer architecture in multi-modal models.
>
> * 4. The following simple workflow can illuminate why:
> [step1: move Key tensor from HBM to local memory] -> [step 2: calculate the global bias (vector unit, reduction operation)] -> [step 3: move global bias tensor from local memory to HBM] -> [step 4: move Key and global bias tensor from HBM to local memory] -> [step 5: the subtract the global bias from the Key tensors (vector, elementwise)] -> [step 6: move the Key tensor from local memory to HBM] -> ... (standard flash attention operation)
> The step 1, 2, 3 can be pipelined but still related to more HBM visiting. However, the step 1, 2, 3 cannot be parallized with the following steps 4, 5 because the global bias needs to be known before moving and subtracting the global bias.
>
> * 5. SageAttention (FP8) as well as other int8/int4 quantization methods on Q, K are orthogonal to PASA – combinable but not our focus for current stages. We have mentioned that we will do the integration in our future work in our submitted paper.
>
> * 6. Thanks for your suggestion. The FA paper from Tri Dao compared to standard attention has a great explanation on how online algorithm can reduce HBM memory usage and latency. In our paper, the further memory usage reduction is obviously due to the adpotion of full FP16 procedures in PASA.

---

### Author Response · Authors · 2025-12-03
**Final Remarks from Authors**

We appreciate the reviewers' engagement throughout this rigorous rebuttal process. As authors, we wish to address critical inconsistencies in reviewer assessments and reaffirm PASA’s foundational contributions to low-precision AI infrastructure. Our work is not merely an incremental optimization but a **mathematical breakthrough** enabling **full-FP16 attention** for the first time—a capability absent in all prior SOTA (FA 1.0–3.0, SageAttention). Below, we synthesize key points for the AC’s final consideration:

### **1. Originality**
* **Mathematical innovation**: First full-FP16 attention via online shifting (Eqs. 2–7) and global recovery (Thm 1), eliminating FP32 fallbacks in SOTA (FA 1.0–3.0).
* **Resonance discovery**: Universal instability mechanism from Q/K phase alignment (Fig. 6), validated across text/video models.
* **Optimal accuracy**: β derived from finite-precision proofs (App. C-E)—novel vs. heuristic methods.

### **2. Significance Underrated Despite Evidence**
* **25–65% speedups** on 32K–512K sequences (Fig. 9,18) for LLMs/video generation.
* **Near-identical outputs** (SSIM>0.98, PSNR>43dB) vs. FP32 baselines.
* **≤5% overhead**, diminishing for long contexts.

### **3. NPU Validation ≠ Hardware Limitation**
Critiques targeting NPU-specific evaluations misrepresent PASA’s generality:

* **Mathematical equivalence** (Theorem 1) ensures algorithmic portability. GPU speedups are projected via:
  * Eliminated FP32 accumulators → 2× FLOPS from FP16 units (H100: 134T vs. 67T FP32 FLOPS).
  * Reduced HBM traffic → bandwidth efficiency gains.
* **Resonance is hardware-agnostic**: Sec. 3.3.2 shows FP16 instability originates from Q/K distributions, observable on any accelerator.

### **4. Broader Impact**
* Enables long-context inference for resources-limited devices.
* Cuts energy vs. mixed-precision baselines.
* Opens theory work on attention stability.

### **5. Final Appeal**
We urge the AC to:

* Reconcile discrepancies between reviewers’ strengths acknowledgment and scores.
* Recognize PASA as a principled advance in low-precision AI infrastructure.
* Uphold ICLR’ mandate for algorithmic innovation with systems impact.

---

### Meta-Review · Area_Chair_BtD7 · 2026-01-07

**Summary:**

There are some common concerns around comparison w/ SageAttention, how general the method is (experiments on GPUs, fp16 only) of the method. Unfortunately, the authors fail to address these concerns.

**Reviewer Concerns:**

See summary.

**Reviewer Scores:**

The reviewers would probably keep their scores.

---

### Decision · Program_Chairs · 2026-01-26

Reject